

# Isotopic Stratification and Non-Equilibrium Processes in a Sub-Arctic Snowpack

Shaakir Shabir Dar[1,2], Eric Klein[3], Pertti Ala-aho[1], Hannu Marttila[1], Sonja Wahl[4,5], Jeffrey Welker[2,6]

[1] Water Energy and Environmental Engineering Research Unit, University of Oulu, Oulu 90570, Finland

[2] Department of Biological Sciences, University of Alaska Anchorage, AK 99508 United States of America.

[3] Department of Geological Sciences, University of Alaska Anchorage, AK 99508, United States of America.

[4] Geophysical Institute, University of Bergen, Bergen 5006, Norway.

[5] Laboratoire des Sciences du Climat et de l'Environnement, Gif sur Yvette 91190, France.

[6] Ecology and Genetics Research Unit, University of Oulu, Oulu 90570, Finland.

*Correspondence to*: Shaakir Shabir Dar (shaakir.dar@oulu.fi)

**Abstract**

Water vapor transport is a primary driver of snowpack metamorphism - a process happening on the microscale which affects important macroscale properties of the snowpack, important for the mass and energy balance of the snowpack. However,

vapor transport and vapor-ice interactions are difficult to observe directly, which necessitates alternative, indirect methods of analysis. Stable water isotopes are an excellent tool to study the vapor-ice continuum in a snowpack because they act as natural tracers of thermodynamics in a snowpack, yet the depth-resolved isotopic signature of pore-space vapor has never been measured in a field setting, leaving a key process gap in snow physics. Here we present the first continuous, multi-level winter record of water-vapor isotopes ($\delta^{18}O$, $\delta^2H$ and *d-excess*) in a snowpack pore space. These data were collected within a

coastal, mid-latitude sub-Arctic snowpack and combined with parallel measurements of overlying air. Coupled with high-resolution meteorological observations and repeated snow-core profiles we explored the temporal and spatial variability of water vapor within the snowpack-atmosphere continuum. Our measurements showed that the pore-space vapor was rarely in isotopic equilibrium with the surrounding ice and that the magnitude and sign of the disequilibrium varied systematically with depth and season. The disequilibrium shifted from diffusion-dominated exchange under cold, strongly stratified

conditions to wind-pumping ventilation during warmer, drier late winter. Late-winter warming, unsaturated atmosphere and stronger winds produced large diurnal swings and midday peaks in water vapor concentration, $\delta^{18}O$, $\delta^2H$ and especially *d-excess* in the snowpack pore space and ambient air, which were indicative of enhanced sublimation and rapid advective mixing. These observations demonstrated that non-equilibrium fractionation by diffusion and wind ventilation reshaped snow and vapor isotopes on hourly to seasonal timescales. This research provides essential insights into how variable

environmental conditions rapidly influenced isotopic variability in the snow-atmosphere continuum, implying that isotope-enabled snow models must capture these rapidly changing processes to avoid biasing hydrological and paleoclimate reconstructions.



## 1 Introduction

Snow is a dynamic force of the polar, subarctic and mountainous landscapes globally that shapes and sustains the region's
hydrological cycles and ecological networks (Cassotta et al., 2022; Nicolaus et al., 2022). Snow plays a crucial role by
influencing climatic and ecological processes (Natali et al., 2019; Pedersen et al., 2021; Welker et al., 2005), including
acting as a vital regulator of the surface energy balance, affecting both the albedo and thermal properties of the environment
(Ala-Aho, Autio, et al., 2021; Biemans et al., 2019; Blume-Werry et al., 2016; Field & Heymsfield, 2015; Niittynen &
Luoto, 2018; Sturm et al., 2017; Thackeray & Fletcher, 2016). As a significant source and reservoir of freshwater, snow
supports various ecosystems during the drier, snow-free seasons (Jespersen et al., 2018; Leffler & Welker, 2013; Meriö et
al., 2019). This seasonal storage and subsequent release of water are essential for sustaining freshwater availability for both
human use and natural ecosystems (Mankin et al., 2015; Rantanen et al., 2022; Siirila-Woodburn et al., 2021). Accelerated
climate change is altering snow cover distribution and variability in multiple ways (Gottlieb & Mankin, 2024; Pulliainen et
al., 2020; Rantanen et al., 2022; Simpkins, 2018). Warmer air raises the rain–snow transition, reduces accumulation, and
triggers mid-winter melts and rain-on-snow events that destabilize and crust the snowpack. Indirectly, thinner or patchy
snow lowers albedo, accelerating soil warming and permafrost thaw, while wind redistribution and vegetation shifts further
modify cover and melt timing. These combined effects advance spring runoff, alter hydrological regimes, and complicate
ecosystem and paleoclimate interpretations. Effective management demands quantifying the mechanisms of snow
metamorphism, ablation and redistribution, and understanding how these rapid cryospheric processes shape ice- and snow-
based climate reconstructions and models.

Recrystallization of snow microstructure through water vapor transport is a key process in Arctic and sub-Arctic snowpacks
which modifies its physical properties (Colbeck, 1983; Jafari et al., 2020, 2022). Driven by the thermodynamic and physical
properties of the ground-snowpack-atmosphere continuum, varying degrees of processes such as, sublimation (Harpold &
Brooks, 2018), deposition (Pinzer & Schneebeli, 2009), redistribution of snow by wind (Walter et al., 2024), water vapor
transport (Jafari et al., 2020, 2022; Jafari & Lehning, 2023), ventilation (Albert, 2002; Ebner et al., 2015; Neumann &
Waddington, 2004) and water vapor exchange between the soil and snowpack (Ala-Aho, Welker, et al., 2021; Domine et al.,
2018), contribute towards snowpack metamorphism. In this context, stable isotopes of hydrogen ($\delta^2$H) and oxygen ($\delta^{18}$O)
serve as powerful integrative markers, capturing the phase changes and transport processes of water within these complex
systems (Beria et al., 2018; Dansgaard, 1964; Galewsky et al., 2016; Gat, 1996). The isotopic signatures, driven by mass and
vapor pressure differences among water isotopologues, during phase transitions and transport can discern the presence and
magnitude of key processes that would otherwise be obscured by the complex interactions within snowpacks (Ala-Aho,
Welker, et al., 2021; Hughes et al., 2021; Sinclair & Marshall, 2008; Wahl et al., 2022).



The post depositional changes in isotopic composition by snow sublimation are predominantly driven by surface and sub-surface processes. In addition to the impact of uneven deposition caused by factors such as precipitation intermittency (Casado et al., 2020; Münch et al., 2017; Persson et al., 2011), seasonal variations in precipitation (Cuffey & Steig, 1998), transport of snow through redistribution by wind (Wahl et al., 2024; Zuhr et al., 2021),  it is being increasingly recognized through laboratory experiments and field observation that the isotopic composition of snow is further influenced by inter-

twined post-depositional processes associated with snow sublimation and vapor transport in snowpacks (Dietrich et al., 2023; Ekaykin et al., 2009; Hughes et al., 2021; Madsen et al., 2019; Neumann et al., 2008; Neumann & Waddington, 2004; Sokratov & Golubev, 2009). Previously, such isotopic fractionation during sublimation, and hence influences of sublimation on the snow isotopic composition, have been disregarded as influential process (Dansgaard, 1964; Friedman, 1991).

At the snowpack surface, water vapor transport encompasses water vapor fluxes between the surface snow to atmosphere and vice versa which dynamically alter the isotopic composition (Hughes et al., 2021; Neumann et al., 2008; Wahl et al., 2022). Sublimation is a non-equilibrium process and has been shown to cause an isotopic enrichment of the remaining snow (Neumann et al., 2008; Sokratov & Golubev, 2009). Surface processes are episodic in nature and are heavily influenced by local atmospheric conditions whereby the rate of surface sublimation (and deposition) is predominantly a function of the

vapor pressure difference between the snow surface and the atmosphere, turbulence and solar radiation (Ala-Aho, Welker, et al., 2021; Gustafson et al., 2010; Harpold & Brooks, 2018; Hughes et al., 2021; Neumann et al., 2008; Sinclair & Marshall, 2008; Wahl et al., 2021)

Below the surface, temperature gradients within the snowpack can drive vapor movements through the interstitial spaces,

leading to isotopic exchanges and recrystallization events (Casado et al., 2021; Touzeau et al., 2018). In Arctic and sub-Arctic seasonal snowpacks, there is a continuous vapor flux from the soil into the basal snow layers, which subsequently lose vapor to upper layers that results in mass and density changes in layers (Calonne et al., 2014; Ebner et al., 2015, 2016; Pinzer & Schneebeli, 2009; Sokratov & Maeno, 2000; Sturm & Benson, 1997). Depending on the thermodynamic properties of ground-snow-atmosphere continuum the layer-to-layer vapor flux occurs from warm to cold regions through diffusion and

convection (Fourteau et al., 2021; Jafari et al., 2020; Jafari & Lehning, 2023; Johnsen et al., 2000; Whillans & Grootes, 1985). However, the isotopic changes throughout the ground-snow-atmosphere continuum due to the vapor transport in snow remains poorly constrained.

Understanding how different vapor-transport processes overlap and affect snowpack isotopic composition is challenging

because current models typically misrepresent water-vapor transport and wrongly assume that sublimation does not fractionate isotopes (Hughes et al. 2021; Wahl et al. 2022, 2024). Moreover, most models impose isotopic equilibrium between snow and pore-space vapor as a boundary condition (Johnsen et al., 2000; Touzeau et al., 2018), which overlooks important non-equilibrium effects. Given the sparse data on stable water isotopes in polar regions, there is a critical need to



expand these datasets to not only refine our process understanding but also to enhance models and parameterizations for
hydrological and paleoclimate applications. Although in situ sampling within the snowpack would capture high-resolution,
sub-daily isotope dynamics, existing vapor isotope measurements (Casado et al., 2016; Hughes et al., 2021; Madsen et al.,
2019; Ritter et al., 2016; Wahl et al., 2021, 2022), have been limited to above-snow sampling, and direct measurements of
water vapor isotopic composition in the snowpack pore space are absent

This study presents the first in situ, multi-level measurements of pore-space water vapor isotopic composition *(δ¹⁸O, δ²H,
and d-excess)* within a snowpack, accompanied by simultaneous measurements of ambient air vapor. These measurements,
integrated with continuous meteorological data and periodic snowpack isotope profiles, are crucial for capturing the dynamic
interactions of water vapor within the snowpack-atmosphere continuum. The aims of this study are to: **(i)** directly observe
and document the depth-dependent isotopic dynamics within the snowpack pore space for the first time, **(ii)** assess the
influence of environmental factors on the variability of pore space water vapor isotopic composition, and **(iii)** analyse the
isotopic fractionation between the snow and the pore space by comparing the equilibrium vapor isotopic composition to the
measured vapor. In addition, the study site, Anchorage, Alaska, represents a distinct setting which is influenced by both
maritime and continental climatic factors (Bailey et al., 2019). This transitional environment is underrepresented in isotope
studies, as much of the focus is often on polar or high-altitude alpine settings.

## 2 Methods

### 2.1 Study Site and Measurement Setup

Measurements were collected between 27 December 2022 to 18 March 2023, at the University of Alaska, Anchorage.
Anchorage (61°N, 149°W, elevation ~55 m) is a coastal city located in south-central Alaska, flanked by the Chugach
Mountains (>3,000 m above sea level). Mean annual total precipitation (rain and snow) in Anchorage is ~2,300 mm, and
approximately 80% falls between November and March 2017 (Bailey et al., 2019).

The measurement profile (Figure 1) was developed to sample air and measure temperature and relative humidity at three
fixed intakes at 5cm, 15cm and 1.5m along the height of a tower. In addition, to these measurements, a soil temperature
sensor and an automated weather station which measures temperature, relative humidity, wind speed and atmospheric
pressure was installed at 1.5m above the ground level. The measurement profile system was setup in the field at the
beginning of winter to allow the installation to be buried under snow precipitation. Additional details about the sensors are in
supplementary S1.



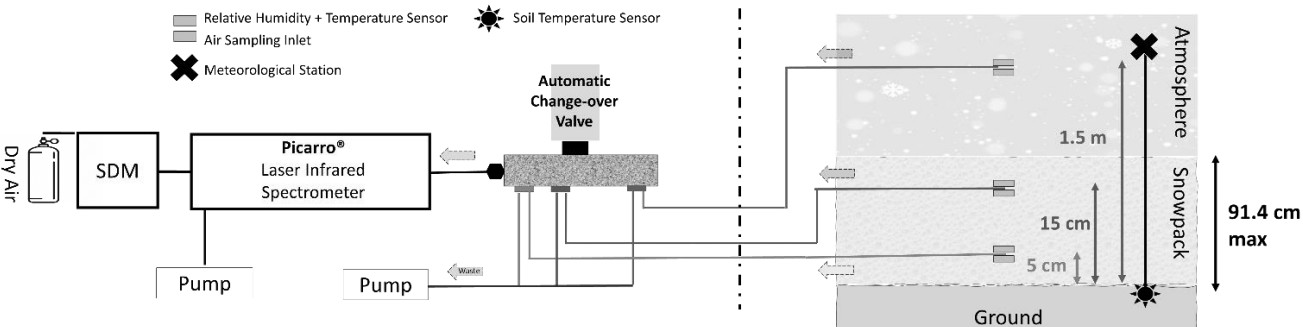

**Figure 1 Schematic diagram of the water vapor isotope measurement setup used at the University of Alaska Anchorage during the 2022–2023 winter season. Air was sampled from three fixed intake heights (5 cm, 15 cm, and 1.5 m above ground level) through heated and insulated tubing connected to a cavity ring-down laser absorption spectrometer (Picarro L2140-i). An automated multi-position valve switched between inlets every 15 minutes. The Standards Delivery Module (SDM) periodically injected reference waters (USGS 45 and USGS 46) for calibration and drift correction. Intakes at 5 cm and 15 cm were located within the snowpack, while the 1.5 m intake sampled ambient air above the snowpack. Temperature and relative humidity sensors were installed at each intake height, along with a soil temperature sensor. An automated weather station at 1.5 m above ground provided atmospheric conditions, including air temperature, relative humidity, wind speed, and atmospheric pressure. The entire system was buried under accumulating snowfall following installation at the beginning of the winter season.**

## 2.2 Water Vapor Isotope Monitoring

A cavity ring-down laser absorption spectrometer (CRDS) (Picarro L2140-i), housed at room temperature, was used to continuously measure the concentration and isotopic composition of water vapor. The vapor was sampled from three fixed intake heights positioned at 5 cm, 15 cm, and 1.5 m along the height of a tower. The length of tubing (6.35 mm outer diameter and 5.8 mm inner diameter) from each sampling point to the analyser inlet was approximately 6.5 m. To prevent condensation and associated isotopic fractionation, the sampling lines were enclosed in double insulation along their entire length and equipped with heating cables, except for the final 0.5 m segment for the inlets located within the snowpack. This unheated portion remained buried, maintaining thermal continuity with the surrounding snow and minimizing the risk of condensation. Automated sequential switching between intake lines was performed every 15 minutes using a Valco Instruments multi-position micro-electric valve actuator. To minimize residence time and prevent carryover between inlet switches, the three inactive sampling lines were continuously flushed by a single pump at a total flow rate of 100 mL min⁻¹ yielding approximately 33 mL min⁻¹ through each inactive line, thus reducing stagnation and ensuring a fresh vapor supply while awaiting measurement. The system included four sampling lines in total, although only data three intake heights (5 cm, 15 cm, and 1.5 m) were used and analysed in this study. The active line connected to the analyser was sampled at the instrument's internal flow rate of approximately 40 mL min⁻¹. To account for potential memory effects after switching heights, measurements collected 8 minutes after and 2 minutes before each valve switch were systematically discarded, based on direct observations of the time required for isotopic readings to stabilize. Following this filtering step, stable 5-minute segments were averaged, resulting in approximately one representative measurement per hour for each intake height. Measurements are reported of volume mixing ratio (ppmv) for humidity and in δ-notation (‰) for oxygen and hydrogen isotopic composition ($\delta^{18}O$, $\delta^{2}H$). Measurements from the 15 cm intake were discarded after 7 March 2023 due to a high-



water concentration alarm triggered by the Picarro L2140-i analyser. The anomaly resulted from water droplets in the sampling line, which may have entered when ambient temperatures rose above freezing and meltwater percolated into the intake. After detecting this, we dried the cavity before resuming measurements at other depths. From that point onward, only data from the 5 cm and 1.5 m intakes were analysed.

Stable isotope ratios were measured approximately every second by the analyser and calibrated using standard protocols
(Aemisegger et al., 2012; Ala-Aho, Welker, et al., 2021; Bailey et al., 2021; Steen-Larsen et al., 2013). The analyser was coupled with a Standards Delivery Module (SDM) for automatic injection of reference waters [USGS 45 ($\delta^{18}O = -2.2‰$, $\delta^2H = -10.3‰$) and USGS 46 ($\delta^{18}O = -29.8‰$, $\delta^2H = -235.8‰$)], to calibrate measurements to the Vienna Standard Mean Ocean Water (VSMOW2) scale. An integrated dry air (HiQ zero air 5.0), system was utilized to measure the vapor streams from the standards, across a range of controlled humidity levels (~700 to ~20,000 ppmv). Each standard was analysed for 12
minutes per humidity level to establish a humidity-isotope response curve. Humidity corrections were applied using a nonlinear regression of the form $\delta X_{corr} = a + {b}/{q}$, where $\delta X$ is the isotopic ratio ($\delta^{18}O$ or $\delta^2H$), $q$ is the water vapor mixing ratio, and $a$ and $b$ are empirically determined constants. This function was optimized for each isotope and applied to all ambient and standard measurements to remove humidity-dependent bias (Figure S1). Standardization to VSMOW2 was performed using repeated injections of USGS 45 and 46 every 48 h, each run lasting 20 minutes. Successful standard runs
were defined as those in which the vapor delivery remained stable, and the injection needle was not blocked by residual deposition. Instrument stability was high throughout the campaign, with standard deviations of 0.21 ‰ (USGS 45) and 0.16 ‰ (USGS 46) for $\delta^{18}O$, and 0.63 ‰ and 1.47 ‰ for $\delta^2H$, respectively, indicating minimal drift over time.

All ambient and standard measurements were corrected for humidity concentration dependence using a non-linear regression
and standardized to the VSMOW2 scale. Based on the uncertainty of both corrections, the measurement accuracy was estimated at ±0.3 ‰ for $\delta^{18}O$ and ±0.6 ‰ for $\delta^2H$. Within the snowpack, most vapor concentration measurements (>90%) at 5 cm and 15 cm depths were above 2,000 ppmv. For ambient vapor, most measurements (85.9%) were above 2000 ppmv, with only 0.33% below 1,000 ppmv. Measurement precision, estimated from the standard deviation of calibration measurements at a constant humidity level, was ±0.4 ‰ and ±0.8 ‰ for $\delta^{18}O$ and $\delta^2H$, respectively, at humidity levels
>2,000 ppmv. Combining accuracy and precision in quadrature, the total measurement uncertainty of water vapor was estimated at ±0.5 ‰ for $\delta^{18}O$ and ±1.0 ‰ for $\delta^2H$.

## 2.3 Snow Sampling and Isotope Analysis

A total of 20 precipitation (snowfall) samples were opportunistically collected in Anchorage between December 2022 and March 2023. Precipitation was collected using a standard cylindrical gauge mounted to a wooden deck at 4 m above ground
level, thereby eliminating the contribution of windblown snow from previous events. Samples were collected at the end of



each precipitation event and typically within 1-3 hr to minimize sublimation. Following sample collection, the gauge was emptied, dried, and reinstalled. Samples were collected from 20 snowfall events out of the 35 days with precipitation recorded. On days with no precipitation, surface snow samples (n=27) were collected. The surface snow (1-2 cm) was scraped directly using a cylindrical vial. Throughout the observation period, 14 snow pits were methodically dug as part of our routine sampling efforts. The sampling protocol was structured according to the snowpack water vapor measurement system's profile, with samples being taken every 5 cm from the ground level to a height of 20 cm. The collected snow samples were, sealed in plastic zip lock bags, and thawed at room temperature. Once thawed, 2 ml of each sample was pipetted into septa-capped glass vials, these samples were subsequently stored at 4 °C. Isotopic analyses of these samples were performed within a two-week timeframe to ensure accuracy and reliability of the data.

The oxygen and hydrogen isotopic compositions of the samples were analysed at the Stable Isotope Laboratory at the University of Alaska Anchorage using a Picarro CRDS system. The setup consisted of an autosampler, a high-precision vaporizer, and a laser-based CRDS analyser. Measurements were calibrated against two international reference standards (USGS45 and USGS46), and an internal laboratory standard (ATW3, Anchorage tap water) was used to monitor instrument performance throughout the analytical runs. All isotopic results are reported in per mil (‰) relative to VSMOW2. Analytical precision was determined from two independent measurement runs, each comprising four replicate injections of an internal laboratory standard (ATW3) and international reference waters (USGS 45 and USGS 46), for a total of eight replicates per standard. The 1σ standard deviations were ±0.03‰ ($\delta^{18}O$) and ±0.16‰ ($\delta^{2}H$) for ATW3, ±0.08‰ ($\delta^{18}O$) and ±1.48‰ ($\delta^{2}H$) for USGS 45, and ±0.04‰ ($\delta^{18}O$) and ±0.59‰ ($\delta^{2}H$) for USGS 46.



# 3 Results

## 3.1 Meteorological and Snow Conditions

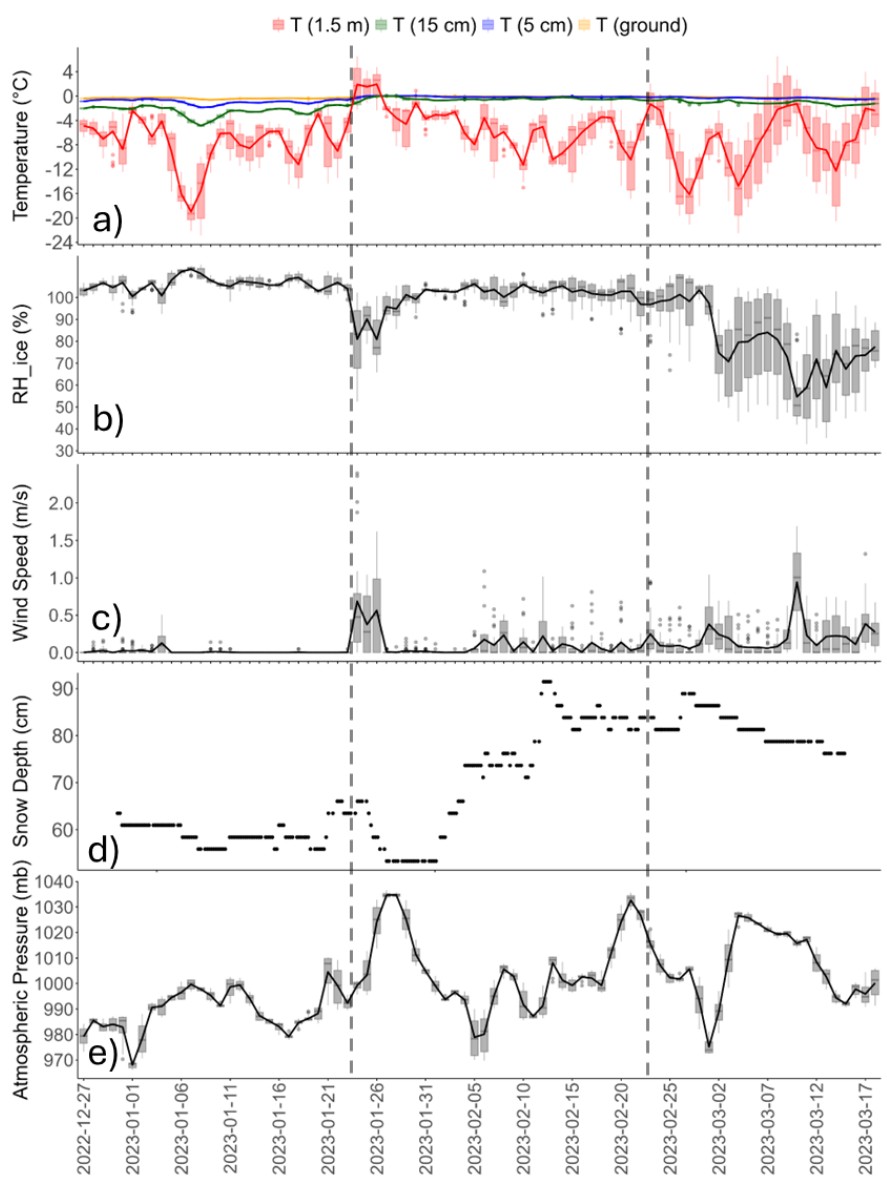

**Figure 2 Environmental data measured at different heights within the ground-snowpack-atmosphere continuum. The bars represent the diurnal variability, and the lines join the daily mean values. a) Color-coded lines represent temperature variations at four depths, soil (orange), snow temperature at 5 cm (blue), snow temperature 15 cm (green), and ambient temperature measured at 1.5 m (red), b) relative humidity calculated with respect to ice, c) wind speed (m/s) at 1.5m, d) snow depth (cm), e) atmospheric pressure (mbar). The dataset is classified into three distinct periods, early, mid, and late, the vertical dashed lines represent the transitions between these periods: January 24, 2023, marks the onset of the first significant warm event, ending the early period. February 23, 2023, marks the beginning of the second warm event, transitioning from the mid period to the late period. The late period includes observations post the second warm event.**





Figure 2 depicts the environmental data collected during the observation period (27 December 2022 to 18 March 2023). Mean hourly air temperature (Ta) was -6.6 ± 4.9 °C, ranging from -22.8 °C to +6.5 °C; the daily maximum of +6.5 °C occurred on 24 January 2023. Temperatures exceeded 0 °C in only 5.6 % of hourly records. Hourly relative humidity (RH) ranged from 33 % to 100 %, averaging 90.2 ± 14.3 %. Relative humidity with respect to ice (RH$_{ice}$), computed with the Goff–

225 Gratch formulation (McDonald, 1965), varied between 33.0 % and 114.9 % and averaged 96.5 ± 16.2 %.

Snow depth was 63.5 cm at the start of the measurements on 27 December 2022. During the 24-26 January warm event it shrank from 66.0 cm to 53.3 cm, then increased to a seasonal maximum of 91.4 cm on 14 February. In the precipitation-free interval 3-18 March depth fell from 86.3 cm to 76.2 cm, coinciding with brief daytime Ta > 0 °C on 7-11 and 17-18 March.

Snowpack temperature at 5 cm above the ground averaged -0.4 ± 0.4 °C (-2.0 °C to 0.1 °C), while at 15 cm they averaged -1.3 ± 1.0 °C (-4.9 °C to 0.02 °C). Relative humidity within the snowpack remained saturated (100 %) at both depths throughout. The warm spell of 24 January 2023 drove the snowpack to isothermal conditions from 26 to 29 January and subsequently reduced the vertical temperature gradient, which nevertheless re-established, though at lower magnitude, once colder weather returned. Soil temperature stayed below freezing for the full record, ranging from -0.6 °C to -0.01 °C and

averaging -0.2 ± 0.1 °C.

The dataset was partitioned into three intervals that capture progressive shifts in atmospheric forcing. The hourly variation in air temperature, relative humidity, and wind speed across the early, mid, and late periods showed distinct trends (Figure S2). The early period (27 December 2022 - 23 January 2023, before the first warm event) featured the lowest and most stable air

temperatures, very high RH, light winds with only slight diurnal variation, and the largest vertical temperature differences within the snowpack. During the mid period (24 January - 22 February 2023), air temperatures became warmer, RH moderated, and winds strengthened modestly; snow and soil temperatures reached their seasonal maxima, yet the snowpack retained small but persistent vertical gradients. The late period (23 February - 18 March 2023) recorded pronounced diurnal swings in, temperature, RH and wind, while vertical snow temperature gradients remained weak. Across all three periods,

soil temperatures consistently exceeded those in the snow layers, and neither soil nor snow temperatures showed any diurnal oscillation.

## 3.2 Diurnal Variations in Water Vapor Concentration and Isotopic Compositions Across Periods and Heights

Figure S3 compiles the mean diurnal cycles of water vapor mixing ratio, $\delta^{18}O$, $\delta^{2}H$, and *d-excess* at the three sampling

heights, while Figure 3 breaks these cycles down by early, mid, and late periods. Throughout the diurnal cycle the snowpack and the overlying air followed the same rhythm, but at different magnitudes. Water vapor mixing ratios inside the snowpack (5 cm and 15 cm) climbed from predawn minima to a broad midday maximum and then declined steadily toward nightfall; ambient air tracked the same shape but remained consistently drier. Isotopically, vapor at every height became enriched (less



negative $\delta^{18}O$ and $\delta^2H$) around midday and depleted overnight. Ambient vapor was always the most enriched and the 5 cm in the snowpack slightly enriched than the 15 cm snowpack layer.  d-excess behaved inversely: snowpack values were lowest at 5 cm, appreciably higher and more variable at 15 cm, while ambient air showed the largest swing rising sharply toward midday and maintaining higher values in the afternoon before tapering off in the evening.

Throughout the winter, the diurnal behaviour of vapor mixing ratio and isotopic composition evolved in step with the early-, mid-, and late-season forcing (Figure 3). The water vapor mixing ratio at 5 cm depth in the early period displayed slight day-night signal; during the mid- and late periods a distinct midday surge developed. At 15 cm the pattern early-period values were slightly higher than at 5 cm but similarly flat; mid-period concentrations matched the 5 cm curve, and by the late period the 15 cm maximum fell below that of the 5 cm layer. Ambient air (1.5 m) always remained drier than the snowpack yet traced a clear diurnal wave in every period; its late-season maxima were the lowest of the three heights and closely approached the 15 cm values.

In terms of vapor isotopic composition, at 5 cm, $\delta^{18}O$ and $\delta^2H$ were steady in the early period, slightly enriched in the mid-period, and depleted again late in the season, while 15 cm vapor was most depleted early on and grew modestly enriched thereafter. From mid-season onward, $\delta^{18}O$ and $\delta^2H$ at 5 cm and 15 cm converged, both layers becoming enriched around local noon. Ambient vapor remained enriched than snowpack vapor throughout; its $\delta^{18}O$ and $\delta^2H$ were most depleted in the early period and displayed pronounced midday enrichment in all periods. Within the snowpack, *d-excess* at 5 cm was consistently lower than at 15 cm and virtually flat during the early and mid-periods; in the late period both depths developed a strong midday enrichment, but night-time values at 15 cm dipped beneath those at 5 cm. Ambient *d-excess* was nearly invariant through early and mid-winter, then adopted a marked diurnal cycle in the late period.





**Figure 3** Plots depicting diurnal variations in water vapor concentration (1) and isotopic compositions ($\delta^{18}O$ (2), $\delta^2H$ (3), d-excess (4)) in ambient air (1.5 m) (a) and at 5cm (c), 15 cm (b) within a snowpack and across three distinct periods: blue-early, yellow-mid and magenta-late.



### 3.3 Isotopic Composition of Water Vapor in the Snowpack-Atmosphere Continuum

Figure 4 presents the time series of vapor $\delta^{18}O$ and *d-excess* at 1.5 m, 15 cm, and 5 cm, while Figure S7 shows the corresponding time series of vapor concentration and isotopic ratios, and Table S1 summarizes the statistics for each period. At both snowpack levels (5 cm and 15 cm) vapor concentrations consistently exceeded those in ambient air, peaking for all heights during the mid-period. Isotopically, the 15 cm level was most depleted in the early-period, whereas vapor at 5 cm stayed comparatively enriched; mid-period vapor at 5 cm and 1.5 m showed a slight enrichment relative to the early and late periods. *d-excess* was highly variable: mean values were largest at 15 cm in the early ($17.76 \pm 2.32$ ‰) and mid ($10.34 \pm 2.97$ ‰) periods and lowest across all heights in the late period.

Figures S5 and S6 present the inter-height correlation analyses of vapor $\delta^{18}O$ and *d-excess* across different periods, with the coefficient of determination ($R^2$) shown to quantify the strength of the linear relationships between measurement levels. For the entire record, ambient vapor $\delta^{18}O$ correlated moderately with the 15 cm level ($R^2 = 0.45$, slope = 0.59) and weakly with 5 cm ($R^2 = 0.10$). The early period showed the tightest coupling between ambient air and 15 cm vapor ($R^2 = 0.78$, slope = 1.08); that link weakened steadily through the mid and late periods. Within the snowpack, the $\delta^{18}O$ relationship between 5 cm and 15 cm strengthened over time, culminating in the late period with $R^2 = 0.77$ and a slope of 1.19, evidence for enhanced internal vapor transport and isotopic exchange as the season progressed. Correlations in *d-excess* were generally weak; nevertheless, the 5 cm-15 cm relationship transitioned from a negative, poorly correlated slope in the early period to a positive, moderately correlated one in the mid period, and by the late period matched the strong $\delta^{18}O$ coupling ($R^2 = 0.77$; slope = 1.19).



**Figure 4** Time series of (a) water-vapor δ¹⁸O and (b) *d-excess* measured at three depths (boxplots: 5 cm in blue, 15 cm in green, 1.5 m in red) alongside discrete precipitation and snow-sample values (points: precipitation in black; surface snow in red; snowpack at 0–5 cm in dark blue, 5–10 cm in light blue, 10–15 cm in dark green, 15–20 cm in light green). Boxplot whiskers show daily variability in vapor isotope composition, and overlaid solid lines trace the daily mean. The vertical dotted lines demarcate early, mid and late periods.



### 3.4 Isotopic Composition of Snowpack and Precipitation

Figure 4 summarizes the temporal variability ($\delta^{18}O$ and *d-excess*) of discrete precipitation and snow-stratigraphy samples. Depth-dependent distributions of $\delta^{18}O$, $\delta^2H$, and *d-excess* within the snowpack are summarized as box-and-whisker plots in Figure S7 and as a continuous depth-versus-isotope heatmap in Figure S8. Precipitation showed the widest spread: $\delta^{18}O$ varied from -28.8 ‰ to -14.2 ‰ (mean -21.2 ‰) by contrast, the layered snowpack displayed a systematic vertical structure that remained far more stable than either precipitation or surface-snow samples ($\delta^{18}O$ = -28.6 ‰ to -15.0 ‰; mean -21.5 ‰).

The vertical profiles of $\delta^{18}O$, $\delta^2H$, and *d-excess* exhibited distinct temporal and depth-dependent patterns throughout the winter (Figure S8). The basal 0-5 cm layer remained consistently enriched in $\delta^{18}O$ and $\delta^2H$, across all sampling dates, with only minor temporal fluctuations and corresponding d-excess values remained low. The 5-10 cm layer showed slightly more variability than the basal layer and *d-excess* in this layer remained mostly stable through January but increased to ~10‰ during late February and March. The 10-15 cm layer exhibited the most depleted values during early and mid-winter with gradual enrichment subsequently. *d-excess* in this layer increased progressively, reaching values above 12‰ by March. In the 15-20 cm layer, $\delta^{18}O$ and $\delta^2H$ were initially moderate but became increasingly enriched through February and March. *d-excess* in the 15-20 cm also rose overtime, peaking at 12-13‰ in March. However, there is uncertainty associated with how spatially representative these values are as records from adjacent snow pits have been shown to be markedly different under the influence of decameter-scale local effects such as wind redeposition of snow, erosion, compaction, and metamorphism (Ekaykin et al., 2014; Petit et al., 1982).

### 3.5 Isotopic Relationships Between $\delta^2H$ and $\delta^{18}O$ in Precipitation, Snow, and Vapor

Figure S9 illustrates the linear regression analyses between $\delta^2H$ and $\delta^{18}O$ for precipitation, snowpack and water vapor. The regression line for precipitation had a slope of 8.3 and an intercept of 15.69, which were both higher than those of the Global Meteoric Water Line (GMWL: $\delta^2H = 8 \times \delta^{18}O + 10$ and the Local Meteoric Water Line (LMWL) specific to Anchorage and south-central Alaska. The LMWL, defined by Bailey et al., (2019), was $\delta^2H = 7.22 \times \delta^{18}O - 11.02$ ($R^2 = 0.94$, n = 332). Using their published dataset, a separate regression for precipitation samples collected during the winter months (December–March) yielded $\delta^2H = 8.06 \times \delta^{18}O + 9.23$ ($R^2 = 0.95$, n = 96).

For surface snow samples collected on precipitation-free days the regression flattened to $\delta^2H = 7.53 \times \delta^{18}O + 4.95$, consistent with kinetic fractionation during sublimation: preferential loss of light isotopes lowers both slope and intercept relative to precipitation. The snow layers displayed varying slopes and intercepts that deviated from both the precipitation and snowpack surface samples. While the regression equations for 0-5 cm and 5-10 cm layers had the lowest slope and intercept values, the 10-15 cm and 15-20 cm layers exhibited steeper slopes and higher intercepts, larger than the precipitation



regression parameters. The $\delta^{18}O$ vs $\delta^2H$ relationship for the 0-5 cm layer had the shallowest slope (5.99) and the most negative intercept (-33.84). The 5-10 cm layer shows a steeper slope (7.26) and a higher intercept than the basal 0-5 cm layer. The 10-15 cm layer had the steepest slope among all the regression lines and the highest intercept (35.98). Similarly, the 15-20 cm layer has a slope of 8.84 and intercept (30.04) which exceeded that of the precipitation $\delta^{18}O$ vs $\delta^2H$ regression parameters.

The $\delta^2H$ vs $\delta^{18}O$ of water vapor demonstrated strong correlations temporally across periods and measurement points. Linear regression lines were fitted to each dataset, and their equations, $R^2$ values, and p-values are displayed in Figure S10. $\delta^2H$–$\delta^{18}O$ correlations in vapor were generally strong ($R^2 > 0.90$) except for 5 cm level in the early period ($R^2=0.76$). Linear regressions yielded intercepts (i.e., d-excess values) that became progressively more negative from the early to late period at both 5 cm and 15 cm within the snowpack

### 3.6 $\delta^{18}O$-d-excess Coupling as a Diagnostic of Kinetic Fractionation in Snow and Vapor

The co-variation of $\delta^{18}O$ and *d-excess* is a sensitive indicator of kinetic processes. A negative slope signals kinetic fractionation: in vapor this reflected mixing with isotopically low sublimated vapor, producing $\delta^{18}O$-depleted, d-excess-enriched air, whereas in snow it recorded post-depositional enrichment of heavy isotopes and concomitant d-excess reduction. Across the full record (Figure S11) $\delta^{18}O$ and *d-excess* of water vapor was negatively correlated in ambient air and in the pore-space at both 5 cm and 15 cm, though the strength of the relation varied. For the integrated dataset the slope was steepest in vapor at 15 cm (-1.67, $R^2 = 0.50$), followed by 5 cm (-1.17, $R^2 = 0.50$) and ambient air (-0.90, $R^2 = 0.26$). The ambient vapor in the mid-period showed the strongest relationship ($R^2 = 0.54$, slope = -1.34). Within the snowpack the slope at both depths became progressively negative from early to late winter.

In the snow layers, the relationship between $\delta^{18}O$ and *d-excess* exhibited a clear depth-dependent trend, that reflected shifts in the dominant processes (Figure S12). In the 0-5 cm basal layer, the relationship was steepest, with a negative slope of -2.01, indicating a strong kinetic control. In the 5–10 cm layer, the slope became less negative (-0.74), suggesting a weakening of kinetic effects, possibly due to reduced vapor exchange with the ground. At 10–15 cm, the slope shifted to positive (1.07), indicating a change in the isotopic regime likely driven by the deposition of upwardly transported vapor, which altered the $\delta^{18}O$–*d-excess* relationship and reduced the signature of kinetic fractionation. Finally, in the 15-20 cm layer, the relationship became statistically non-significant ($R^2 = 0.16$), suggesting further decoupling of $\delta^{18}O$ and *d-excess*, potentially due to isotopic homogenization, mixing with ambient vapor, or weakened vertical vapor fluxes that reduced coherent isotopic gradients.





**3.7 Physical Controls: Temperature and Humidity Effects**

Linear-regression analyses linking vapor isotopic composition to physical drivers, ambient meteorological variables (air temperature, relative humidity) and internal snowpack conditions (vertical temperature gradients) are presented in Figures S13-S15 and Figure 5. Across the full record (Figure S13) ambient air temperature explained a moderate fraction of vapor $\delta^{18}O$ variance in the overlying atmosphere ($\delta^{18}O = -28.91 + 0.40\ T$; $R^2 = 0.43$). This temperature sensitivity attenuated with depth in the snowpack, falling to $R^2 = 0.38$ at 15 cm ($\delta^{18}O = -32.83 + 0.35\ T$) and to $R^2 = 0.26$ at 5 cm ($\delta^{18}O = -32.05 + 0.22\ T$). A temporal breakdown revealed a monotonic decline in temperature control through the season: during the early period the ambient slope was 0.39 ‰ °C$^{-1}$ with $R^2 = 0.84$ (Figure S13), dropped to 0.21 ‰ °C$^{-1}$ with $R^2 = 0.22$ by late winter (Figure S13), and showed a parallel weakening at 15 cm, while the 5 cm level remained poorly correlated throughout.

In Figure 5 (left panel), we examined the relationship between the layer-to-layer temperature gradient ($\Delta T = T_{15} - T_5$) and the 380  $\delta^{18}O$ difference ($\Delta\delta^{18}O$) for snow and vapor. When all snow samples were included, a linear regression yielded a weak, non-significant positive trend. The two red-circled points were sampled days with the highest and lowest ambient air temperature observations; they fall outside the main cluster and disproportionately increase the apparent spread (or variance) of the dataset. After excluding these two extreme-temperature outliers, the remaining snow samples produced a much stronger correlation: $\Delta\delta^{18}O_{(snow,\ filtered)} = 4.64\ \Delta T + 2.35$ ($R^2 = 0.65$, $p = 0.001$). In this filtered dataset, the slope increased nearly 385  fourfold and $\Delta T$ explained roughly 65 % of the variance in $\Delta\delta^{18}O_{(snow)}$. The regression intercept shifted from +0.84 ‰ (full dataset) to +2.36 ‰ once extremes were removed, indicating a baseline isotopic offset between the two layers at $\Delta T = 0$.

The corresponding pore-space vapor data (Figure 5) showed a clearer dependence on $\Delta T$. For vapor measurements, the regression was $\Delta\delta^{18}O_{(vapor)} = 2.38\ \Delta T + 0.22$ ($R^2 = 0.467$, $p < 0.001$), indicating a significant positive correlation: as the 390  temperature gradient increased, the upper-layer vapor became depleted in $^{18}O$ relative to the deeper layer. The intercept of +0.22 ‰ indicates a small but consistent upper-layer depletion at $\Delta T = 0$ in the filtered vapor dataset. The d-excess difference ($\Delta$d-excess) between 15 cm and 5 cm against the same $\Delta T$ is shown in Figure 5. For snow, there was no meaningful relationship, $\Delta$d-excess $_{(snow)}$ remained effectively constant regardless of $\Delta T$. For the pore-space vapor, a strong negative relationship: $\Delta$d-excess$_{(vapor)} = -8.16\ \Delta T - 1.72$ ($R^2 = 0.62$, $p < 0.001$). This fit shows that approximately 62 % of 395  the variability in $\Delta$d-excess$_{(vapor)}$ can be explained by $\Delta T$, with d-excess differences declining sharply as $\Delta T$ increases toward zero.





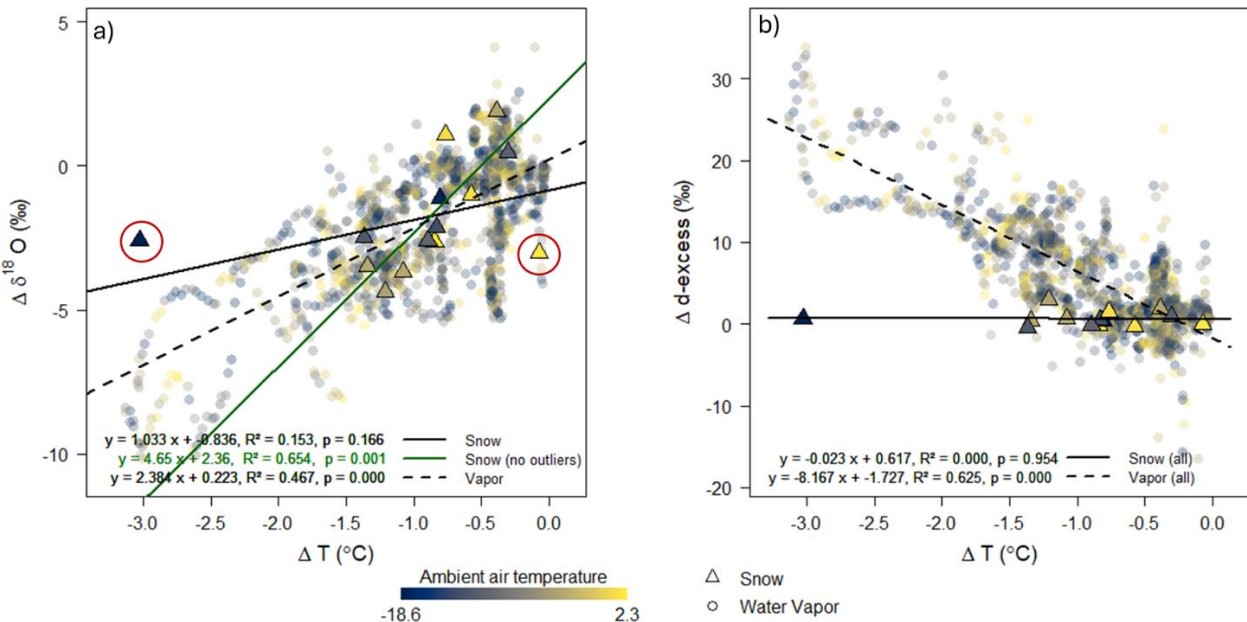

**Figure 5 Coupling between the vertical temperature gradient in the snowpack and isotopic contrasts in pore-space vapor. Two-**
400 **panel comparison of isotopic differences $\Delta\delta^{18}$O (‰) (a) and $\Delta$d-excess (‰) (b) versus temperature gradients ($\Delta$T = $T_{15\,cm}$ − $T_{5\,cm}$) in**
**the snowpack, with each point shaded according to ambient air temperature for snow (triangles) and vapor (solid circles). Two**
**outliers (January 8 and January 26, 2023) are highlighted with red circles. Three regression lines are shown: a solid black line for**
**all snow data, a solid green line for snow after removing outliers, and a dashed black line for vapor.**

Humidity control emerged most clearly under unsaturated conditions (Figure S14). In the early and mid-periods, $RH_{ice}$
hovered near 100 %, ambient *d-excess* showed no systematic response. During the late period, however, $RH_{ice}$ depressions
produced the theoretically expected negative trend: *d-excess* = 23.91 − 0.22 $RH_{ice}$ ($R^2$ = 0.41), which steepened to -
0.28 ‰ %$^{-1}$ and $R^2$ = 0.50 when analysis was restricted to $RH_{ice}$ < 100 %. These fits demonstrated that kinetic enrichment in
d-excess was strongly modulated by atmospheric saturation deficit once the air column dried. Temperature also modulated
the vapor d-excess, but with a different seasonal fingerprint (Fig. S15). Ambient air displayed its strongest anticorrelation
with temperature in mid-winter, whereas the early and late periods were weak. Inside the snowpack, the early period *d-*
*excess* increased with temperature at 5 cm (slope 0.73 ‰ °C$^{-1}$, $R^2$ = 0.42), while at 15 cm the relationship reversed (-
0.25 ‰ °C$^{-1}$, $R^2$ = 0.23). Both relations decayed as the season progressed, suggesting that ventilation and phase changes
increasingly outweighed pure temperature control. To isolate purely unsaturated conditions, we examined the precipitation-
415 free interval 2-7 March 2023 (Figure 6). Across all three heights *d-excess* declined with $RH_{ice}$. Ambient vapor attained values
up to +20 ‰, whereas pore vapor spanned -20 ‰ to +10 ‰, implicating a ground-vapor contribution. Scatter increased with
wind speed, particularly inside the snowpack, implying that mechanical ventilation modulated snow-vapor exchange rates.




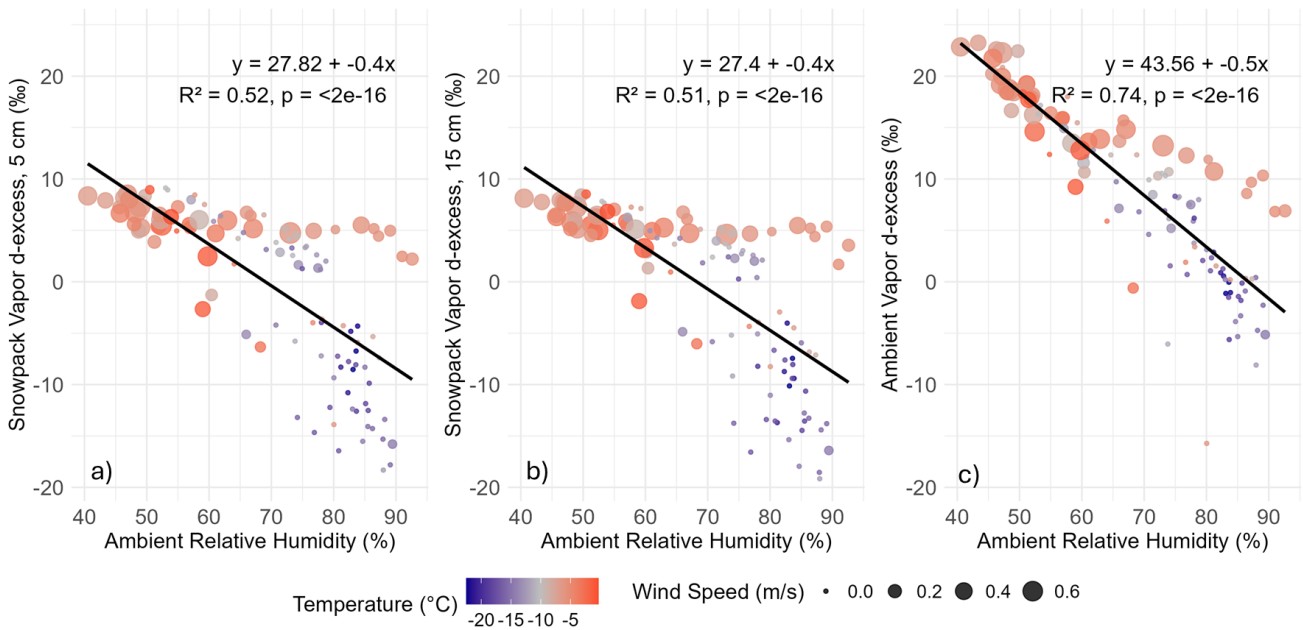

**Figure 6 Linear regression relationship between *d-excess* of snowpack vapor and ambient relative humidity at different heights on during precipitation free days of March (2 March 2023 to 7 March 2023). The three panels represent snowpack vapor at 5 cm (a), 15 cm (b), and ambient vapor at 1.5 m (c). Each point represents an hourly observation, with point size corresponding to wind speed and colour indicating ambient temperature.**





## 3.8 Temperature Influences on Isotopic Disequilibrium and Vapor-Snow Interactions in the Snowpack

**Figure 7 Time series plots displaying the *δ¹⁸O* (a) and *d-excess* (b) of observed snowpack vapor and isotopic composition of vapor calculated assuming equilibrium with snow layers at 4 depths 0-5 cm, 5-10 cm, 10-15 cm and 15-20 cm). The equilibrium vapor isotopic composition was calculated using the formulation by** (Ellehoj et al., 2013)**.**

430



Figure 7 shows the measured snowpack vapor with the theoretical equilibrium vapor calculated from the isotopic composition of the underlying snow at four depth intervals (0-5, 5-10, 10-15, 15-20 cm) using equilibrium fractionation factors by Ellehoj et al., (2013). The measured snowpack vapor $\delta^{18}O$ and *d-excess* fluctuated much more than their equilibrium counterparts because of small fluctuations in snowpack temperature and snow-layer isotopic composition. The measured $\delta^{18}O$ disequilibria (Figure S16), were predominantly positive and spread over a broad range, indicating frequent enrichment relative to equilibrium. In contrast, the d-excess disequilibria (Figure S16) remained consistently below zero and the disequilibrium got stronger as the season progressed, in line with sublimation effects. Across snow depths, a clear pattern emerged especially in $\delta^{18}O$ disequilibrium. The deepest basal layers (0-5 cm and 5-10 cm) exhibited relatively muted isotopic shifts: their $\delta^{18}O$ offsets were smaller in magnitude and varied little through time. By contrast, the middle layer (10-15 cm) $\delta^{18}O$ offsets showed the widest scatter. Across all depths $\delta^{18}O$ offsets showed a strong skew toward enrichment. In Figure S17, the disequilibrium is plotted against ambient temperature. The basal layers (0-10 cm) show weak, non-significant trends (~ 0.18 ‰ °C$^{-1}$; R$^2 \leq 0.25$) (Figure S17a). In contrast, the 10-15 cm and 15-20 cm layers exhibited steep, significant positive slopes (~ 0.4 ‰ °C$^{-1}$; R$^2 \sim 0.5$) (Figure S17b). *d-excess* disequilibrium, however, showed no temperature signal at any depth (not plotted).

## 4 Discussion

### 4.1 Meteorology and Diurnal Vapor Dynamics

The environmental conditions during the observation period played a key role in shaping the isotopic composition of snow and water vapor in the snowpack-atmosphere system. The distinct early (27 Dec 2022 - 23 Jan 2023), mid (24 Jan - 22 Feb 2023), and late (23 Feb - 18 Mar 2023) periods corresponded to transitions in temperature, humidity, and wind speed (Figure 2), which directly influenced water vapor transport, and post-depositional isotope modifications. In the early period, low ambient temperatures, saturated atmosphere and extended night led to a shallow, stably stratified boundary layer with limited turbulent exchange. Under these conditions, the snowpack maintained a temperature and isotopic stratification. As the season progressed, increasing air temperatures, unsaturated atmosphere and higher wind speeds facilitated greater turbulent exchange, internal snowpack mixing, and ventilation in the mid and late periods.

Our results revealed distinct diurnal signatures in concentration $\delta^{18}O$, $\delta^2H$ and *d-excess* of water vapor at different snowpack heights (5 cm, 15 cm, and 1.5 m), with variations across the early, mid, and late periods (Figure 3). Water vapor concentrations at all heights exhibited a clear diurnal cycle, peaking at midday, then declining toward evening, although amplitude grew from early to late winter. The isotopic composition of water vapor ($\delta^{18}O$, $\delta^2H$) follows a similar diurnal cycle, becoming more enriched (less negative) at midday and depleted (more negative) overnight and in the early morning. This midday enrichment in heavy isotopes was strongest in the ambient air (1.5 m) in the late period and to a lesser extent in the snowpack vapor. This indicated that the surface snow underwent kinetic fractionation due to increased sublimation rates





in warmer conditions, but this enrichment primarily reflected the mixing of relatively enriched sublimation-derived vapor with the advected vapor. The vapor produced from this process, although depleted relative to the snow source itself, was isotopically enriched compared to ambient atmospheric vapor and resulted in an observed increase in ambient vapor $\delta^{18}O$ and $\delta^2H$ at midday. This means that the snow was driving the local vapor isotopic composition on the daily timescales. These

diurnal vapor cycles were broadly consistent with Arctic summer observations from Greenland (Klein et al., 2015; Madsen et al., 2019; Steen-Larsen et al., 2013), and Antarctica (e.g. Dutrievoz et al., (2025)) where daytime sublimation enriched near-surface vapor in heavy isotopes via kinetic fractionation. Conversely, when deposition was the dominant process, water vapor $\delta^{18}O$ and $\delta^2H$ became significantly depleted (Ritter et al., 2016). Model results from (Wahl et al., 2021) also showed vapor isotope cycles in seasonal snow that track diurnal thermal gradients, reinforcing that surface energy fluxes were key

drivers of isotopic variability.

### 4.2 Physical Drivers: Meteorological Controls and Internal Thermal Gradients

#### 4.2.1 Temperature Controls on $\delta^{18}O$ in Ambient and Snowpack Vapor

The isotopic composition of water vapor was closely coupled with ambient meteorological conditions, notably temperature and relative humidity, as well as internal snowpack temperature gradients. In the overlying atmosphere, ambient vapor $\delta^{18}O$

was strongly correlated with air temperature during the early winter period, reflecting a near-equilibrium response to radiative cooling and stable boundary-layer dynamics (Figure S13). However, this relationship weakened during the mid and late periods as sublimation fluxes increased, relative humidity declined, and kinetic effects, such as fractionation under drier, windier conditions, became more prominent. In the snowpack interior, a similar pattern was observed where the $\delta^{18}O$-$T$ relationship weakened from early to late winter. At 5 cm, however, the $\delta^{18}O$-$T$ relationship was markedly weaker across all

periods, with shallow slopes (~0.16–0.22 ‰ °C$^{-1}$). This suggests that vapor at 5 cm was increasingly decoupled from ambient atmospheric conditions and influenced by subsurface processes at the ground-snowpack interface than by direct meteorological forcing. Notably, during the early period, $\delta^{18}O$ at 15 cm was consistently more depleted than in ambient air, Within the snowpack, the 15 cm layer also exhibited a significant $\delta^{18}O$-$T$ relationship during early winter (slope = 0.47 ‰ °C$^{-1}$), slightly steeper than in the ambient air (slope = 0.39 ‰ °C$^{-1}$) and consistent depletion in $\delta^{18}O$ relative to both 5 cm

and ambient air, indicating kinetic fractionation due to vapor being transported upward through diffusion under a temperature gradient, carrying a depleted isotopic signal.

#### 4.2.2 Layer-to-Layer Isotopic Responses in Snow and Pore-Space Vapor Under Thermal Gradients

The results in Figure 5 revealed strikingly different behaviours for snow-layer and pore-space vapor isotopes under varying

internal temperature gradients. First, snow $\Delta\delta^{18}O$ was only a reliable proxy for $\Delta T$ when sampling excluded extreme thermal events, surface warming or cooling disrupted and masked the underlying trend. Second, continuous vapor measurements are





essential for capturing the true kinetics of isotopic exchange: pore-space vapor $\delta^{18}O$ and *d-excess* both responded linearly and predictably to $\Delta T$, whereas snow samples, by virtue of being bulk, infrequent, and daytime-biased, cannot resolve rapid changes or preserve transient kinetic signals. Third, snow $\Delta$d-excess remained uncorrelated with $\Delta T$ because bulk sampling

masked micro-scale isotopic variations at crystal edges, whereas pore-space vapor retained that real-time signature. Finally, the persistent positive intercept in snow $\Delta\delta^{18}O$ at $\Delta T = 0$ suggested that layering and metamorphism left a lasting isotopic signature that was independent of the instantaneous thermal gradient. Thus, internal snowpack temperature gradients, primarily control deep-snowpack vapor transport and the associated diffusive isotopic fractionation processes.

Temperature gradients have long been identified as key drivers of vapor transport and isotopic fractionation within snowpacks. Jafari et al. (2020) simulated diffusive vapor transport under thermal gradients and found that, in shallow Arctic snowpacks, diffusive vapor fluxes can reduce density by up to ~60 kg m$^{-3}$ in bottom layers, effectively thickening a dense layer aloft. Although that study focused on density, it implies vigorous upward vapor transport under stratified conditions, which inevitably carries a distinct isotopic signature. Likewise, Johnsen et al. (2000) and Touzeau et al., (2018)

demonstrated that vapor diffusion driven by temperature and isotopic gradients continuously modifies the isotopic composition of firn and ice cores. Our in-situ observations at centimetre resolution confirm and quantify this behaviour: even modest thermal gradients ($|\Delta T| \sim 1$ °C) produced multi-per-mille shifts in both $\delta^{18}O$ and d-excess of snow and pore space vapor.

### 4.2.3 Humidity and Kinetic Fractionation: *d-excess* as a Diagnostic

Relative humidity also exerted a strong control on ambient vapor *d-excess* indicative of enhanced kinetic fractionation under unsaturated conditions (Figure S14). This relationship was further supported by analysis during precipitation-free intervals, which consistently showed that unsaturated air resulted in higher *d-excess* values of water vapor (Figure 6). During these dry, unsaturated conditions, the snow surface $\delta^{18}O$ became progressively enriched, shifting from approximately -24‰ to -

20.5‰, accompanied by a decrease in *d-excess* from about 14.6‰ to 3.3‰. Field studies in Greenland have revealed strong day-night cycles in snow isotopic composition driven by vapor-snow exchange. For example, Madsen et al. (2019) found that clear-sky summer days induced significant diurnal swings in the upper sub-centimetre of snow. Similarly, Hughes et al. (2021) conducted laboratory and summer-field experiments in NE Greenland, showing that continuous sublimation enriched the very top snow layer by up to ~8 ‰ in $\delta^{18}O$ (with decreased d-excess) over a few days. Wahl et al. 2022 found that the

snow surface isotope variability could be explained by 30-50% when considering surface humidity fluxes.



### 4.2.4 Wind Pumping and Midday Enrichment of d-excess

During late winter we observed pronounced, transient d-excess peaks, inside the snowpack, and the overlying atmosphere, occurred each day around local noon (Figure 3). This contrasted sharply with the relatively stable, diffusion-dominated d-
excess values recorded in early winter. This indicated a transition in vapor transport dynamics, driven by surface warming, decreasing RH, and enhanced wind speeds (Figure S2). Midday warming favored surface sublimation, thereby increasing the ambient vapor in d-excess. Concurrently, elevated wind speeds enhanced ventilation of the snowpack through pressure-driven airflow ("wind pumping"), which enabled rapid mixing and redistribution of this enriched vapor within the pore spaces. The result was a sharp increase in snowpack vapor *d-excess* (to ~+10‰), approaching ambient vapor values (up to
+20‰), while vertical $\delta^{18}O$ gradients flattened, signaling effective pore-space mixing (Figure 6). These likely reflected combined processes: **(i)** vapor loss from the snowpack surface during sublimation raised the *d-excess* of ambient vapor locally, and **(ii)** intrusion of *d-excess* enriched ambient vapor into the snowpack via ventilation. Under calm and humid conditions, snowpack vapor remained diffusion-dominated, especially at 5 cm and 15 cm, with *d-excess* values clustering near 0‰ to -10‰, indicative of ground-sourced vapor with diffusive imprint.


The rapid propagation of this isotopic signal down to 5 cm, coupled with the collapse of vertical $\delta^{18}O$ gradients, mirrors the laboratory findings of Ebner et al., (2017). Their wind tunnel experiments demonstrated that forced vapor flow with a contrasting isotopic signature can measurably alter snow $\delta^{18}O$ within 24 to 84 hours, depending on temperature gradients and microstructural controls. These findings resonate strongly with the conceptual framework of Neumann & Waddington,
(2004), who modeled how advection-driven vapor transport superimposed on diffusion can alter isotopic distributions within firn and Town et al., (2008), who used a Rayleigh-fractionation framework to simulate 3-7 ‰ annual $\delta^{18}O$ enrichment under 5-10 m s$^{-1}$ winds, emphasizing that forced ventilation, pore-space diffusion, and intra-ice-grain diffusion can substantially modify near-surface snow on ice-sheets. Their simulations highlighted how wind pumping not only disrupted internal isotopic gradients but also reduced the time required for isotopic signals to propagate from the atmosphere into the
snowpack. Seasonal snowpacks have much higher porosity and air permeability likely amplifying wind pumping effects, making isotopic modifications more pronounced. Taken together, these field observations and complementary experimental and modelling studies underscore that seasonal snowpacks are highly dynamic isotope reactors: diurnal ventilation events can imprint atmospheric signatures onto buried layers within hours, with important consequences for interpreting paleoclimate records if wind-driven advection is not explicitly parameterized.


### 4.3 Vertical Isotope Patterns and Post-Depositional Fractionation Processes in Seasonal Snowpack Stratigraphy

The vertical isotope profiles revealed a well-structured snowpack shaped by surface–atmosphere exchange, internal vapor redistribution, and ground vapor influence. Although precipitation during the study period exhibited substantial isotopic



variability, driven by shifting synoptic patterns such as cold northerly intrusions and warm, south-westerly Aleutian Low
events (Bailey et al., 2019), these initial signals were rapidly modified once snow was incorporated into the snowpack. The
snowpack stratigraphy showed a structured and temporally evolving isotopic profile. Basal enrichment (0-5 cm) remained
stable, shaped by soil vapor diffusion (Ala-Aho, Welker, et al., 2021; Friedman, 1991), while the 5-10 cm layer showed
slight $\delta^{18}O$ variability and a delayed *d-excess* increase. The 10-15 cm layer, although buried under 40-60cm of snow, showed
variations, initially the most depleted, became enriched over time, and the upper 15-20 cm showed progressive enrichment
and rising d-excess by late winter. These patterns reflected internal vapor redistribution and surface exchange, consistent
with previous Arctic snowpack studies highlighting post-depositional isotopic modification (Ala-Aho et al., 2021).

The depth-dependent $\delta^{18}O$-$\delta^2H$ (Figure S10) and $\delta^{18}O$-*d-excess* relationships for snow and vapor, provided key insights into
the dominant fractionation processes that shaped snowpack and water vapor isotopic evolution. At the snowpack surface,
lower slopes, and intercepts of $\delta^{18}O$-$\delta^2H$ relationship compared to precipitation suggested that sublimation preferentially
removes lighter isotopes, enriching the remaining snow in heavier isotopes. In the basal layers (0-5 cm and 5-10 cm), where
the shallowest slopes and lowest intercepts were observed, indicated basal sublimation and ground vapor influence which
enhanced kinetic fractionation. In contrast, the steepest slopes and highest intercepts occurred in the 10-15 cm and 15-20 cm
layers, suggesting that vapor deposition played a more significant role at these depths. For water vapor, the $\delta^{18}O$-$\delta^2H$
relationship evolved temporally, reflecting increasing kinetic fractionation over the course of the season. The intercepts of
the regression equations became progressively more negative (Figure S11), indicating the increasing contribution of
isotopically depleted sublimated vapor to the overall water vapor pool.

The negative correlation between $\delta^{18}O$ and *d-excess* across all datasets reinforced the role of kinetic fractionation during
sublimation (Figure S11 and Figure S12), although such a relationship is generally expected under non-equilibrium
processes. Slope differences between snow and vapor highlighted their contrasting sensitivities: snow recorded localized
surface and ground conditions, while vapor integrated broader pore-space dynamics. The relatively stable isotope structure
of snow layers, paired with the persistently dynamic nature of vapor, emphasized the differential response of the two phases
to environmental forcing.

**4.4 Isotopic Disequilibrium and Vapor-Snow Interactions**

Our observations revealed a pervasive disequilibrium between measured pore-space vapor and the theoretical equilibrium
vapor composition (Figure 7 and Figure S16). The disequilibrium displayed systematic trends with depth: near the base they
were relatively tightly clustered, while in the upper layers both the spread of values and the prominence of extreme
deviations grew. The $\delta^{18}O$ disequilibrium metrics were all shifted above zero and showed greater right skew and broader tails
in the upper layers, indicating more frequent and larger enrichment events higher in the snowpack. Conversely, the d-excess
disequilibrium metrics were negative on average but also became increasingly variable and right-skewed toward the snow



surface, reflecting stronger episodic departures from equilibrium in the mid- to upper columns. The comparison between measured vapor isotopic composition and equilibrium vapor highlights the depth-dependent controls on snowpack isotopic evolution, driven by ground influence, diffusion, and ventilation.


When we regressed $\Delta\delta^{18}O$ (measured - equilibrium) against ambient air temperature, only the mid and upper layers exhibited a significant positive slope, whereas the basal 0-10 cm layers showed weak, non-significant trends (Figure S17). No layer showed any meaningful temperature dependence in d-excess disequilibrium. This depth-stratified pattern suggests that ambient air temperature likely affected isotopic disequilibrium in the snowpack indirectly, by strengthening/weakening

thermal gradients (and thus diffusion) under cold, calm conditions, and by enhancing ventilation (via lower RH and higher winds) during warmer periods. Before the January 24 warm event, when strong internal temperature gradients prevailed, and winds were light the $\delta^{18}O$ and *d-excess* disequilibria at 15 cm were not driven by direct exchange with the overlying air but by upward diffusion of vapor from the basal layers. The fact that $\delta^{18}O$ at 5 cm exceeded both the equilibrium vapor and $\delta^{18}O$ at 15 cm, pointed to a significant soil-vapor input and upward molecular vapor diffusion through the snowpack, in line with

previous studies (Ebner et al., 2016; Ebner et al., 2017; Ala-Aho et al., 2021; Friedman et al., 1991; Sinclair & Marshall, 2008). In early winter, the snowpack behaved as a closed system, developing strong vertical isotopic stratification. Under these stratified conditions, kinetic diffusion also controlled *d-excess*, yielding the highest vapor *d-excess* and largest disequilibrium in the mid-snowpack (around 10-15 cm).

Following the warm event, internal thermal gradients collapsed, and ventilation became the primary transport mode as warmer, windier conditions enhanced sublimation and wind-pumping, boosting advective exchange. Despite this homogenization, systematic $\Delta\delta^{18}O$ differences persisted between 5 cm and 15 cm because they draw on different vapor endmembers: 5 cm remained anchored to ground-sourced vapor (enriched in $^{18}O$, low in d-excess), whereas 15 cm was increasingly influenced by ventilated ambient vapor. Crucially, the disequilibrium calculated against the local snow $\delta^{18}O$; an

enriched snow layer and hence enriched equilibrium vapor, making measured $\Delta\delta^{18}O$ appear smaller, and vice versa. Thus, even fully mixed vapor can show depth-dependent disequilibrium simply due to how the equilibrium reference shifts with snow stratigraphy. In sum, temperature while not a direct isotopic driver serves as a powerful diagnostic of the transition from diffusion-dominated basal conditions to ventilation-dominated mid/upper snowpack exchange.

To bring together these multilayered observations, we now present a conceptual schematic (Figure 8) that distils the dominant vapor-transport and isotope-fractionation pathways operating in our seasonal snowpack.





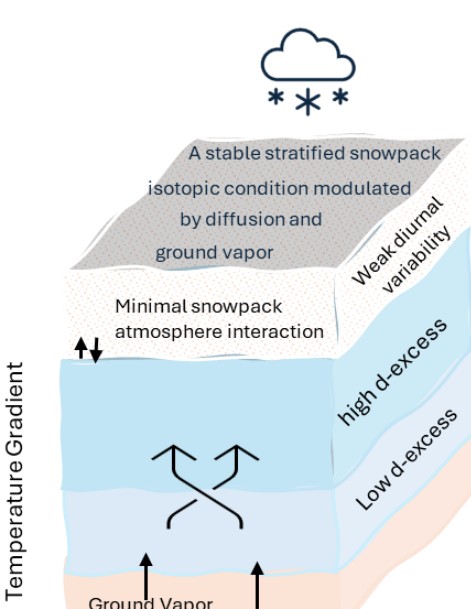
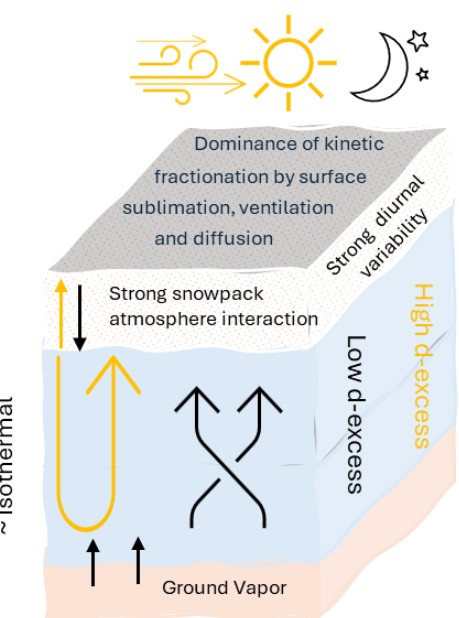

**Figure 8 Conceptual schematic of snowpack vapor-transport regimes and isotopic responses Left (Early-winter, diffusion-dominated): Under cold, calm conditions with strong internal thermal gradients and weak boundary-layer exchange, vapor migrates upward primarily by molecular diffusion causing in the upper layers with high d-excess. Vertical isotopic stratification persists, and diurnal variability remains muted. Right (Late-winter, ventilation by day (yellow) / diffusion at night (black)): During warm, low-RH afternoons, surface sublimation and pressure-driven "wind pumping" flush the mid/upper snowpack with enriched vapor, flattening $\delta^{18}O$ gradients and producing sharp midday *d-excess* peaks. Nocturnal cooling and reduced winds restore diffusion-controlled exchange and vapor deposition, overall, the snowpack remains isotopically uniform than in early winter.**

## 4.5 Limitations of the Study

While this study provides valuable insights into the isotopic dynamics of water vapor within seasonal snowpacks, several limitations should be considered when interpreting the results. First, measurements were taken along a single vertical profile, preventing assessment of spatial variability across the snowfield. Small-scale heterogeneity, such as surface undulations, drifts, or local compaction, may drive differing vapor and isotope gradients in nearby locations. Second, although vapor was sampled at 5 cm and 15 cm depths, the 15 cm inlet was unintentionally buried well below its intended mid-snowpack position during an early-season snowstorm. To preserve the integrity of the snowpack structure, the inlet was not repositioned, highlighting the potential value of deploying evenly spaced inlets throughout the snowpack depth to better capture evolving isotopic stratigraphy. Third, although the forced-air inlet system supported high-frequency measurements, it may have disturbed local vapor pressures and pore-space equilibrium. However, the small volume of vapor extracted relative to total pore-space vapor produced no systematic artifacts. For future deployments, the use of diffusion–dilution sampling (Volkmann & Weiler, 2014) could be explored to minimize perturbations. Fourth, all snow pit samples were collected during daylight hours, limiting our ability to examine nocturnal vapor transport and condensation processes under colder,



nonequilibrium conditions. Finally, although key environmental variables were monitored, solar radiation, snow surface
temperature, and internal snow structure (e.g. porosity, density) were not directly measured. Incorporating these parameters,
as well as eddy covariance flux data, would help bridge microscale vapor–snow interactions with macroscale energy and
mass fluxes. Future studies that combine denser vertical sampling (in soil, snowpack and atmosphere), continuous radiative
and surface flux monitoring, and spatial replication across snow-covered areas will help generalize these findings and better
constrain ground-snow-atmosphere vapor exchange processes.

**5 Conclusions and Implications for Further Research**

We presented the first continuous, winter-long measurements of pore-space vapor isotopic composition ($\delta^{18}O$, $\delta^2H$, and d-
excess) in a coastal, mid-latitude sub-Arctic snowpack overlying permeable soil. It lays the foundation to use snowpack
vapor isotopes to improve the understanding of snowpack metamorphism which has previously only been done in laboratory
settings (Ebner et al., 2017). These data constituted the inaugural record of pore-vapor isotopes and revealed dynamic
variability on both daily and seasonal timescales. In early winter, under cold, calm, and strongly stratified conditions, vapor
exchange was governed primarily by molecular diffusion. By late winter, however, rising temperatures, declining humidity,
and increasing wind speeds triggered wind-pumping-driven ventilation. This shift flattened vertical $\delta^{18}O$ gradients,
introduced ambient vapor with elevated d-excess, and generated pronounced diurnal isotopic cycles in both the atmosphere
and snowpack pore-space vapor, cycles that coincided with late-period diurnal oscillations in temperature, relative humidity,
and wind speed.

Most isotope-enabled climate and snowpack models still treat deposited snow as an isotopically inert layer or allow only
slow diffusive exchange. Our high-frequency winter record shows that sublimation, diffusion, and wind-pumping can
reshape both snow and pore-vapor isotopes within hours to days, implying that models must incorporate non-equilibrium
fractionation and seasonally shifting transport regimes. Because kinetic overprints can raise $\delta^{18}O$, depress or spike d-excess,
and blur layer-to-layer signals, ice-core and snow-pit records may carry systematic post-depositional biases. Variability in
ice core d-excess, often interpreted solely as changes in moisture source, can equally reflect year-to-year differences in
sublimation intensity or ventilation. Robust paleoclimate reconstructions therefore require snowpack models that simulate
these active processes, not just initial precipitation isotopes.


Soil-derived vapor is a major end-member in terrestrial snow, fractionation schemes calibrated on ice-sheet surfaces cannot
be transferred uncritically to land snowpacks. Isotope-enabled snow models must therefore (i) couple vapor transport to both
the atmosphere and the soil, (ii) include parameterizations for isotopic imprints of water vapor transport (e.g. diffusion,
ventilation, and convection), (iii) include non-equilibrium fractionation during sublimation and deposition, and (iv) account
for evolving porosity and permeability as snow densifies.

During firnification, a process marked by increasing density, reduced pore space, and altered vapor transport pathways, these same kinetic and transport processes will change, affecting how isotopic signals are preserved or modified. Our data set provides a benchmark for calibrating such models: features such as midday *d-excess* peaks, rapid δ swings, depth-dependent

fractionation, and their evolution through densification should be reproducible in any physically based simulation. However, because firn transformations alter pore-space structure, directly connecting these snowpack observations to ice-core records requires additional work to quantify how isotopic signals are modified during compaction.

Future research should focus on how variations in snow layering and density affect isotopic gradients. Targeted sampling of

distinct density horizons (e.g., surface hoar, crusts, depth hoar, ice lenses) with sub-daily isotope measurements would reveal how localized porosity and permeability control vapor fluxes and non-equilibrium fractionation. Laboratory experiments and numerical simulations could then explore how evolving density during firnification shifts the balance between diffusion and advection, shaping signal preservation. By explicitly linking density-driven pore geometry changes to isotopic evolution, these efforts will improve parameterizations in isotope-enabled snowpack and ice-core models, reducing uncertainty in

paleoclimate reconstructions and hydrological forecasts.

**Data availability**. The data presented in this paper are available at https://doi.org/10.5281/zenodo.15630146 with a restricted access. The data will be made available to reviewers upon request and made open to the public after publication.

**Supplement.** The supplement related to this article is available on-line.

**Author contributions.** SSD and JW conceptualised and designed the experiments. SSD did the formal analysis, created the visualisations and wrote of the original draft. ESK, PA, HM, SW and JMW all contributed to the data interpretations and edited the manuscript.


**Competing interests**. None of the authors have any competing interests.

**Acknowledgements**. Shaakir Shabir Dar acknowledges funding from United States India Educational Foundation, Fulbright-Nehru Fellowship (Award no. 2682/FNPDR/2021) and Marie Skłodowska-Curie Actions European Postdoctoral

Fellowship (Award no. 101062626, iSUBLIME-HORIZON-MSCA-2021-PF-01). Pertti Ala-aho was supported by Research Council of Finland, SNOMLT project (Award no. 347348). Sonja Wahl acknowledges funding from the Norwegian Research Council, SnowDOGS project (Award no. 335140). Jeffrey M Welker acknowledges funding from his UArctic Research Chairship, Research Council of Finland (Award no. 368772) and National Science Foundation, USA (Award no.



2133156). We thank John Ferguson for help with sample analysis at the Stable Isotope Lab of the University of Alaska,
Anchorage and Valtteri Hyöky for help with vapor isotope data calibration.

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
