# Peer review of "Isotopic Stratification and Non-Equilibrium Processes in a Sub-Arctic Snowpack"

_EGUsphere, 2025_

## Referee Comment (RC1)

**Review of EGU-2025-2724: "Isotopic Stratification and Non-Equilibrium Processes in a Sub-Arctic Snowpack"**

This paper reports the first continuous measurements of isotopic vapor composition performed in a natural sub-arctic snowpack, alongside measurements of the meteorological and isotopic compositions of the overlying air and ice within the snowpack. The benefits of these measurements are two-fold: (i) they can be used as discriminants to disentangle and clarify the physical processes at play in snowpacks (e.g. vapor diffusive versus advective transport or isotopic (non)-equilibrium between the air and ice phases) and (ii) they document important post-depositional effects in snowpack that alter cryospheric isotopic records and hydrological interpretations.

I found the experimental method and its description compelling. The obtained data are very valuable and of great help to further our understanding of vapor physics in snowpacks. The authors provide an extensive analysis of the data and put forward a sound interpretation in terms of processes at plays. The main drawback I could see is that due to the large amount of information, it is sometimes difficult to identify the salient points of this study. Personally, the major information I got from the article (besides the demonstration that vapor isotopic concentration can be continuously monitored) are: (i) vapor and ice in snow are in thermodynamical disequilibrium (contrary to what is usually assumed by snow/firn isotopic models) (ii) the soil appears have a detectable influence on the isotopic composition of the bottom snow, (iii) vapor isotopic composition shows snowpack-wide mixing events during high wind events, indicative of wind-induced ventilation, and (iv) there is a strong interaction between the vapor composition of the near surface snow and that of the overlying atmosphere (consistently with previous results reported in the literature). But streamlining the presentation of this amount of information is not an easy task and to be frank I do not have that many ideas on how to do it. In all cases, I think the complexity of the observed data (with correlations that strengthen/weaken over time) shows that the isotopic profiles (vapor and ice) in the snowpack result from the interaction of multiple processes, and provide a nice benchmark to evaluate theoretical and numerical models.

In conclusion, I think this study is well suited for the cryosphere and I recommend its publication after some minor revisions.

**General Comments**

**Influence of pumping on water vapor measurements:** The water vapor to be analyzed by the CRDS is sampled by pumping through 6.5m of tubing. I was of the general impression that the pumping of a gas mixture can introduce some "pressure-gradient fractionation". Has it been quantified and could it have a detectable influence on the measurements performed in this study or is the effect too small to affect the results?

**Characterization of the soil:** Has the ice/vapor in the underlying soil been isotopically characterized? Or at least, do the measurements in the bottom part of the snowpack give a consistent picture of the isotopic nature of the soil vapor (enriched $\delta^{18}O$ and low d-excess if I follow **L613**)? This seems quite important to me in order to be able to discuss the effects of water vapor flux from entering the snowpack from the ground.

 I did not manage to access the restricted data. If I understand correctly, they contain all relevant processed data discussed in the text (meteorological data, snowpack samples data, vapor composition data, precipitation/surface composition data, and snow temperature/humidity data). I trust the authors to have included all that (and maybe some more that I'm forgetting), but I just wanted to be sure.

**Specific Comments**

**L46** - As far as I understand the impacts of snow decrease on soil temperature cannot be simplified to a consistent acceleration of soil warming. Thinner snow covers also favor the cooling of the permafrost during winter. The net impact of a changing snow cover on the ground can either be a warming or cooling effect, depending on the particular conditions at play (e.g. Lawrence and Slater, 2010).

**L65** - I would not use "predominantly" here, as I do not understand what could be a third type of process that is neither a surface nor a sub-surface process.

**L95 and elsewhere** - For me snow is the resulting (macroscopic) mixture of ice + humid air. Therefore, the notion of (dis)-equilibrium between the air and the snow sounds a bit odd. I understand that sometimes "snow" is used interchangeably with "ice", but since this paper focuses on the distinguishing the isotopic composition of the air phase to that of the ice, it might be worth to use the word "ice" rather than "snow" when specifically discussing the composition of the ice matrix.

**L171** - I do not understand how the calibration and correction of the measurements was performed. The text implies that there is only one a and b per isotopes (so one {a,b} for DHO and another {a,b} for $H_2^{18}O$), but I do not see how it could be used to get from a raw δX to a corrected δX$_{corr}$. Perhaps there is a missing δX in the formula.

**L173** - Why isn't the correction applied to snowpack vapor measurements as well?

**L180** - What are the two corrections? There is one for humidity but it is not clear to me what is the other one.

**L231** - I just wanted to mention that the fact that the relative humidity of the vapor was measured within the snow during the whole season and consistently shows saturation is very valuable. It is a crucial piece of information for vapor physics dynamics.

**L235** - If possible please include a brief description of the snowpack stratigraphy. It doesn't have to be super precise I think, but the potential presence (or absence) of slabs or crusts that impede water vapor movement could be important information.

**L280** - Perhaps rename the section to precise that it focuses on variations at the seasonal time scale, when the previous section focused on diurnal variations.

**L295** - I understand the strengthening of the correlation between the 5 and 15cm levels could suggest some sort of enhanced mixing in the later period (I'm thinking of wind pumping based on the latter part of the article). But isn't the argument partly countered by the fact that

the correlation between the ambient air and the snowpack diminishes over time, that could suggest reduced mixing between the snowpack and the ambient air?

**L479** - Could you explain and/or specify why the relation between ambient $\delta^{18}O$ and T in the early period indicates near-equilibrium? The equilibrium fractionation factor decreases with temperature. I would thus expect a negative correlation between $\delta^{18}O$ and T if the ambient vapor was in isotopic equilibrium with some ice source. But perhaps the authors meant something else.

**L618** - I do not follow why temperature serves as a diagnostic of the diffusion-advection transition. Could you elaborate?

**Technical Comments**

**L123** - What is the depth of the ground sensor?

**Fig 1.** - If possible, it might be interesting to put a picture of the actual setup. It could be done in the supplementary material not to clutter the main part of the article.

**L151** - Is there a reason why the fourth line was not included in the study in the end?

**L171** - Just out of curiosity, is there a theoretical reason to search the correction under this specific form, or it is based on the shape of the curves in Fig. S1?

**L171** - The function yields $\delta X_{corr}$ but only $\delta X$ is presented instead in the text.

**L179** - It seems that this sentence just restates what has been said in the paragraph above (standards and ambient measurements are corrected with some non-linear function of q).

**L223 and Fig. 2** - Precise that this is the relative humidity of the overlying atmosphere (not to be confused with the relative humidity of the snow vapor)

**L255 and L629** - I think the word "more" is missing.

**L380** - From what I understand, $\Delta\delta^{18}O = \delta^{18}O_{15} - \delta^{18}O_{5}$ (the difference in isotopic composition between spatial points in the snowpack, which can be applied both for the ice matrix and the vapor). If so, please indicate it clearly so it is not confused with the ice-vapor isotopic difference.

**L379 to 396** - The split between the two paragraphs is a bit strange to me. It is §{$\Delta\delta^{18}O$ in ice} then §{$\Delta\delta^{18}O$ in vapor + $\Delta$d-excess in ice and vapor}. I would rather expect §{$\Delta\delta^{18}O$ in ice and in vapor} and then §{$\Delta$d-excess in ice and vapor}.

**L497** - I think there are $\Delta$ missing: isn't it the $\Delta\delta^{18}O$ and $\Delta$d-excess that respond to $\Delta T$?

**L535** - It seems that the last part of the paragraph is a just a re-wording of what has been stated above (namely that surface sublimation release high d-excess vapor, that then

increase the d-excess of the ambient air and of the deeper snow when wind pumping is present). Consider removing it to lighten the text.

**L596** - $\Delta\delta^{18}O$ is already defined as the difference in $\delta^{18}O$ between layers. Perhaps simply use the name "$^{18}O$ disequilibrium (defined as measured $\delta^{18}O$ minus the theoretical $\delta^{18}O$ equilibrium value)".

**Supplementary Material**
**L3** - Remove "com".

---

## Referee Comment (RC2)

**Review of EGU-2025-2724: "Isotopic Stratification and Non-Equilibrium Processes in a Sub-Arctic Snowpack"**

This study presents the first continuous field measurements of water vapor isotopes inside a snowpack, along with air and snow core data influenced by both maritime and continental climatic factors. The results show that vapor inside the snowpack is usually not in equilibrium with the surrounding ice, and this imbalance changes with depth and season. The author concluded that in cold, stable conditions, diffusion controls vapor exchange, while in warmer, windier late-winter conditions, ventilation by wind becomes more important. These late-winter conditions also caused strong daily swings in vapor and isotopic values, linked to sublimation and rapid mixing with the atmosphere.

The authors provide an extensive analysis of the data and provided some detailed interpretation on processes affecting the isotopic signal inside the snowpack. The findings highlight that snowpack isotopes are strongly influenced by short-term environmental changes, and models must include these processes to avoid errors in hydrology and climate reconstructions. However, I have some drawbacks about the correctness of the measurement and the results of the vapor inside the snowpack or I misinterpreted the data wrong, see my comments below.

In conclusion, this paper is well written and the subject is appropriate for Cryosphere but I recommend major revisions to improve the reliability of the data.

**Major comment:**

I'm a bit concerned about the correctness of the measurement looking at the data in Figure 2 and 3. Your temperature measurement inside the snowpack (Figure 2) shows a minimum of around -4 degrees (corresponding to a saturation vapor concentration of around 4300 ppmv) but your concentration measurements with the Picarro (Figure 3) show a minimum of around 2700 ppmv. This would indicate that you have a relative humidity of only 65% inside the pore space of the snow pack which is not possible. Do you have an explanation for this?

The same for the ambient measurement, you measured the lowest temperature at 1.5m of

around -20 degrees (corresponding to a saturation vapor concentration of around 1500 ppmv) but your lowest measured concentration was around 2500 ppmv. This would indicate that your ambient air was sometimes oversaturated. Or could it be that your Picarro sucked in ice/water particles or is the Campbell Scientific AP200 intake also used for the isotope vapor? Maybe you could provide a picture of the system in the supplement materials.

You claimed that the pore spaces of the snowpack was at saturation but I could not find any evidence in the data. Therefore, it seems like that either your setup was not leak-tight or there is an offset in the humidity measurement of your Picarro.

**General comments:**

- Sometimes you are talking about that vapor measurements were taken at 5 or 15 cm depth. This is a bit misleading because when you are talking about the depth some readers will see it as the distance from the surface into the snowpack. I would suggest to skip the 'depth' or call it 'height' to improve the readiness.
- I'm missing important information to get a better understanding of the setup and to interpret the results. How did you check that your system is leak-tight and did you perform a humidity correction of the Picarro?
- I'm a bit concern about your comparison of your results between the three different periods (early, mid and late) because the snow depth is not constant and is changing up to 30 cm during the campaign. I would suggest to provide an explanation why you can still compare the different periods (especially about the data of the two intakes within the snowpack) with each other.
- How do you justify that 15 cm is a mid-snowpack position for a snowpack depth of 90 cm?
- I questioning the wind pumping effect on the isotope signal without knowing your spatial density profile of your snowpack. In addition, based on Figure 2 you measured a max. wind speed of 2 m/s and you are intakes are between 0.4m and 0.8m below the snowpack surface. For this condition I would not expect any significant ventilation inside the snow pack (see Colbeck et al., 1989). But without knowing your density profile of your snowpack it is hard to make a conclusion.

**Specific comments:**

2) Methods

- Line 115: Did you check your system for leaks? And did you do a humidity correction of the Picarro? It seems like that either your setup was not leak-tight or there is an offset in the humidity measurement of your Picarro. See major comments.
- Line 115: In Figure 2 you showed the snow depth. I would suggest to quickly mention it here with all the other parameters.

- Line 115: Is there a specific reason why you measured the isotopic composition at 5cm, 15cm and 1.5m above ground? I would suggest to provide a reason for it.
- Line 135-136: How did you make sure that the sensors did not get frozen during the measurements, especially your humidity sensor?
- Line 141-142: How did you make sure that the intakes are not sucking in small ice/water particles?
- Line 145-147: Could you provide a bit more explanation how you buried the last 0.5m inside the snow. Were the last 0.5m directly horizontally inside the snow? If not, how did you make sure that the temperature gradient inside the snowpack does not have an impact on condensation inside the tube?
- Line 146: '... and minimizing the risk of condensation': It is hard to understand what you want to say here. I assume the reason why you didn't heat the last 0.5m is because you could have potential heating up/melting of the snow at the intake. Why is there then a risk of condensation? Could you elaborate a bit more.
- Line 148-149: How far was the valve actuator away from the intakes?
- Line 149: '... the three inactive sampling lines...' How can you have three inactive sampling lines when you have only three sampling lines. Why do you have a fourth one? Is it relevant to have a fourth one? Please provide more information.
- Line 149: '... continuously flushed': Please specific mention it that it continuously flush the line with the vapor of the pore space (I assume this is what you did).
- Line 154: I don't see a reason why you should discharge the last 2 minutes before switching the valve. Did you see any impact on the signal in the last 2 minutes? If yes, why do you think that it has to do something with the valve? Could you elaborate a bit more?
- Line 155-156: I would delete this sentences because it confuses a bit. Actually, you measured one intake twice during one hour.
- Line 158-161: I'm questioning your explanation about water percolation. Looking at Figure 2 I don't see a reason why you should have water percolation into the snowpack. The ambient temperature was far below zero and also the temperature at 15cm shows value below zero. Could it not be that the Picarro sucked in ice crystals which melted in the heated tube? How do you prevent that ice particles can enter the intakes?
- Line 170: 'Humidity corrections were applied...': I assume it is not a humidity correction but a humidity-isotope correction.
- Line 177: '... minimal drift over time': At what humidity level did you perform your drift measurement? Close to the humidity levels of the intakes inside the snowpack?
- Line 181-182: You indicate that 10% of your vapor concentration measurements inside the snow were below 2000 ppmv. I doubt that this is realistic for a snowpack where the lowest temperature was around -4 degrees. Could you check what was the snowpack temperature when you measured vapor concentration below 2000 ppmv.
- Line 195-196: Could you elaborate a bit more why you didn't include sample measuring above 20 cm from ground level? Such measurements would help to get a better idea

how the signal changed with depth and why there was a potential disequilibrium at the intakes.

- Line 202: '... using a Picarro CRDS system.' Could you mention which model e.g. L2120, L2130 or L2140?
- Line 216: I assume that the relative humidity was measured at 1.5 m. Please mention it.
- Line 231: 'Relative humidity within the snowpack remained saturated ...': I can't find any evidence that your data shows saturated conditions inside the snowpack. See comments above.

3) Results

- Figure 2: How did you measure the snow depth and what was the resolution of it?
- Figure 3: Could you also add the temperature profile to see whether the concentration (ppmv) is following the temperature profile or not.
- Figure 3: Is it possible to extract a time-shift between the measured atmosphere data and inside the two snowpack locations? If yes, would it be possible to compare this time-shift with the diffusion time ($\Delta t = L^2/D$) from the snowpack surface to the intake locations (maybe include a tortuosity factor for the diffusion length inside the snowpack) to check whether it is consistent.
- Figure 4: I would suggest to change the colour of the '0-5 cm' and 5-10 cm' snow data points. It is hard to distinguish it.
- Figure S4: How do you explain that you measured a vapor concentration below 1900 ppmv inside the snow pack but your temperature is only around -4 degrees? I'm also surprised that the vapor concentration inside the snowpack is almost the same as the 1.5m measurement.

4) Discussion

- Line 459-460: '... at different snowpack heights (5 cm, 15 cm, and 1.5 m)...' -> please rewrite this part because 1.5m does not belong to the snowpack but to the atmosphere.
- Line 487-488: '... than in ambient air, Within the ...': I assume that the sentence ends after 'ambient air.'
- Line 527: I questioning this paragraph without knowing your spatial density profile of your snowpack. In addition, based on Figure 2 you measured a max. wind speed of 2 m/s and you are intakes are between 0.4m and 0.8m below the snowpack surface. For this condition I would not expect any significant ventilation inside the snow pack (see Colbeck et al., 1989). But without knowing your density profile of your snowpack it is hard to make a conclusion. I would suggest that you provide more evidence to support your hypothesis. Maybe you could provide an estimation about what wind speed inside the snowpack would be needed to transport the atmospheric vapor into the snowpack. E.g you could try to extract a time-shift between the measured atmosphere data and inside the two snowpack locations and calculate a wind speed needed to transport the signal into the snowpack.

- Line 608: '... in the mid-snowpack (around 10-15 cm).': How do you justify that 15 cm is a mid-snowpack position for a snowpack depth of 90 cm?
- Line 606 and 610: Would it be possible to provide an explanation that first 'In early winter, the snowpack behaved as a closed system...' and afterwards the snowpack is not closed anymore and wind-pumping and '... ventilation became the primary transport mode...'? Looking at your snow depth data on Figure 2 I would expect that your two intakes locations inside the snowpack are even more decoupled from the atmosphere because the snowpack is rising by additional 20-30cm.
- Line 636-637: Could you elaborate this a bit more? What do you mean '... buried well below its intended mid-snowpack position during an early-season snowstorm'? Didn't you want to keep the intake locations constant at 5 cm and 15 cm above ground? Or what was your intended mid-snowpack position? And how do you justify that 15 cm is a mid-snowpack position for a snowpack depth of 90 cm?

5) Conclusion

- Line 692: The link is not working. Please correct it.

S1) Supplement material

- Line 4: '... fluctuations in water vapor com concentrations.' -> remove 'com'
- Line 7: '... relative humidity at the four inlets, a HOBO...': I think there is a type. You are talking about four inlets but based on your experimental setup you have only three (5cm, 15cm and 1.5m).

**Technical comments:**

- Line 255: '... the snowpack slightly enriched than the ...' Typo -> '... the snowpack was slightly more enriched than the ...'
- Line 339: 'The 5-10 cm layer shows a steeper slope ...' Typo -> 'The 5-10 cm layer showed a steeper slope ...'
- Line 354: '... vapor was negatively correlated ...' Typo -> '... vapor were negatively correlated ...'
- Line 678: 'Our data set ...' -> 'Our dataset ...'
- Remove redundant commas in citations (e.g., "Bailey et al., (2019)" → "Bailey et al. (2019)")

---

## Referee Comment (RC3)

**Review of manuscript "Isotopic Stratification and Non-Equilibrium Processes in a sub-Arctic Snowpack**

**Authors: Dar et al.**

**Submitted to Cryosphere.**

The manuscript argues that diffusion and wind ventilation cause non-equilibrium fractionation, which reshapes the water isotopic composition within the snowpack in both the vapor and the snow on hourly to seasonal timescales.

The argument is based on a comprehensive dataset of water vapor isotope observations from both above and within the snowpack, combined with snowpack isotope profiles and direct temperature measurements.

The authors support their conclusions based on the finding that their measurements of the water vapor isotopic composition show that the pore-space vapor is rarely in isotopic equilibrium with the surrounding ice.

Their experimental setup consists of two inlets, located respectively 5 and 15 cm above the ground, which are buried by snow deposited throughout the season. At the beginning of the campaign, the 15 cm inlet is 45 cm below the snow surface, while at the end of the campaign, it is 65 cm below the snow surface.

The target question of the manuscript is crucial for understanding the physical processes that affect the climate signal recorded in the stable isotopic composition stored in the ice crystals that comprise the snowpack. Since the early work of Waddington et al., who hypothesized that wind pumping is important for driving the isotopic composition in the snow, discussions and attempts have been made to quantify the vapor transport within the snowpack through direct measurements.

Unfortunately, it is this reviewer's view that the authors make similar mistakes as previous attempts to measure interstitial water vapor, in that they disregard the influence of their measurements on the medium they are trying to measure, i.e., the interstitial water vapor.

Contrary to the statement on line 640, "However, the small volume of vapor extracted relative to total pore-space vapor produced no systematic artifacts," I will argue below that

the volume of vapor extracted in fact is producing artifacts, which prevent the authors from reaching a robust conclusion.

The fundamental problem with their setup is that the authors remove air from the snowpack continuously:

L 149 "The three inactive sampling lines were continuously flushed by a single pump at a total flow rate of 100 ml/min, yielding approximately 33 ml/min through each inactive line"

L152 "The active line connected to the analyzer was sampled at the instrument´s internal flow rate of approximately 40 ml/min.

This means that a constant flow through each inlet line buried in the snow is 33-40 ml/min. As the air for the inlet lines in the snowpack cannot come from the ground, it must come from the atmosphere above the snowpack. This means that a total of 66-80 sccm of air will flow to the combined inlets in the snow. See Figure 1 below as a sketch of the setup.

[Figure]

*Figure 1: Sketch of setup*

The fundamental question that then arises is, where does the air come from? It must be such that the air will follow a path of the smallest integrated resistance. This means that the air cannot come from infinity. On the other side, it also seems unlikely that the air follows a tube flow straight from the surface through a path with a diameter of 6 mm equal to the inlet ID.

Below I therefore consider two situations:

Conservative option 1: The air enters the snowpack through an area with a diameter of twice the depth of the inlet (90 cm beginning of season and 130 cm end of season) and travels through a semi-sphere of the snowpack. See figure 2 for a sketch of the setup.

[Figure]

*Figure 2: Sketch of a conservative thought experiment of how the air travels through the snowpack.*

Making the following assumption that the density of the snow is 500 kg/m3 based on the information in the text that the authors observed compaction during periods of no precipitation and the relatively high temperatures at the field site.

For the calculations of snow depth at 60 cm:

The volume of the semi-sphere is 190e3 cm$^3$. This means that the volume of air is 95e3 cm$^3$. With a flow rate of respectively 33 or 40 sccm per inlet, this would mean that all the interstitial air will be replaced every 20 to 24 hours.

Even for this relatively conservative estimate, I will therefore argue that the author's argument on line 640, "However, the small volume of vapor extracted relative to total pore-space vapor produced no systematic artifacts," does not hold.

A more realistic flow field through the snowpack would probably be better described by a column flow with a diameter of 40 cm from the top of the snowpack down to the inlet. Figure 3 illustrates perhaps a more realistic flow field through the snowpack.

[Figure]

*Figure 3: Illustration of a perhaps more realistic flow field through snowpack.*

Again, carrying out the calculations for a snow depth of 60 cm.

In this case, the volume of the column is 56e3 cm$^3$ and the total volume of the interstitial air is 28 cm$^3$. For the given flow rates through the inlet lines, this would mean that the air in the snow column is replaced every 6 to 7 hours.

Based on these calculations, I believe that the effect on the interstitial vapor isotopic composition, which the authors attribute to diffusion and wind ventilation, is a result of the forced transport of atmospheric air through the snowpack.

Further support for my conclusion is provided by the authors in the observed snowpack temperature at the depth of the inlet and the observed diurnal variations in humidity of the interstitial vapor:

Figure S2 shows that no diurnal variation in snowpack temperature is observed at the 5 and 15 cm inlet

[Figure]

**Figure S2** Plots depicting diurnal patterns in ambient air temperature (C), relative humidity w.r.t. ice (%), wind speed (m/s), and soil and snow temperatures (5cm and 15cm) depth, measured over three distinct periods: early, mid, and late.

However, the authors also demonstrate clear diurnal variations in the observed humidity at the inlets, as shown in Figure 3.

[Figure]

**Figure 3** Plots depicting diurnal variations in water vapor concentration (1) and isotopic compositions ($\delta^{18}O$ (2), $\delta^2H$ (3), d-excess (4)) in ambient air (1.5 m) (a) and at 5cm (c), 15 cm (b) within a snowpack and across three distinct periods: blue-early, yellow-mid and magenta-late.

Following the findings of Neumann et al. 2009 (Sublimation rate and the mass-transfer coefficient for snow sublimation): *"Our data (e.g. Fig. 4) suggest that the snow sublimates*

*rapidly, and that for snow samples with thickness 1 cm or greater, pore spaces in snow are typically saturated with vapor, as other investigators have assumed [5]"*

This means that the snow at the 5 and 15 cm inlet must have a diurnal variation in temperature in order to create a diurnal variation in humidity. As the temperature at 5 and 15 cm does not show any diurnal temperature variation, this means that the flow of air into the inlets must influence the temperature in the vicinity of the inlets, but not be recorded by the temperature observations. Hence, one might conclude that even the "*perhaps more realistic flow field in the snowpack*" is in fact still too conservative, and the flow from the surface to the inlet follows an even narrower corridor through the snowpack.

Based on my above argument, I therefore believe that the setup in the manuscript of Dar et al. does not allow the authors to reach the presented conclusions.

---

## Author Comment (AC1)

**Response to Reviewer #1 for EGU-2025-2724:** *Isotopic Stratification and Non-Equilibrium Processes in a Sub-Arctic Snowpack*

We thank the reviewer for their thoughtful and constructive comments on our manuscript. We are pleased that they found the dataset and analysis valuable, and we appreciate their recommendation for publication after minor revisions. We have carefully addressed all comments and added details where requested.

For clarity, the reviewer's original comments are shown in *italics*, followed by our responses in **blue plain text**.

*This paper reports the first continuous measurements of isotopic vapor composition performed in a natural sub-arctic snowpack, alongside measurements of the meteorological and isotopic compositions of the overlying air and ice within the snowpack. The benefits of these measurements are two-fold: (i) they can be used as discriminants to disentangle and clarify the physical processes at play in snowpacks (e.g. vapor diffusive versus advective transport or isotopic (non)-equilibrium between the air and ice phases) and (ii) they document important post-depositional effects in snowpack that alter cryospheric isotopic records and hydrological interpretations. I found the experimental method and its description compelling. The obtained data are very valuable and of great help to further our understanding of vapor physics in snowpacks. The authors provide an extensive analysis of the data and put forward a sound interpretation in terms of processes at plays. The main drawback I could see is that due to the large amount of information, it is sometimes difficult to identify the salient points of this study. Personally, the major information I got from the article (besides the demonstration that vapor isotopic concentration can be continuously monitored) are: (i) vapor and ice in snow are in thermodynamical disequilibrium (contrary to what is usually assumed by snow/firn isotopic models) (ii) the soil appears have a detectable influence on the isotopic composition of the bottom snow, (iii) vapor isotopic composition shows snowpack-wide mixing events during high wind events, indicative of wind-induced ventilation, and (iv) there is a strong interaction between the vapor composition of the near surface snow and that of the overlying atmosphere (consistently with previous results reported in the literature). But streamlining the presentation of this amount of information is not an easy task and to be frank I do not have that many ideas on how to do it. In all cases, I think the complexity of the observed data (with correlations that strengthen/weaken over time) shows that the isotopic profiles (vapor and ice) in the snowpack result from the interaction of multiple processes and provide a nice benchmark to evaluate theoretical and numerical models. In conclusion, I think this study is well suited for the cryosphere and I recommend its publication after some minor revisions.*

**General Comments**

*Influence of pumping on water vapor measurements: The water vapor to be analyzed by the CRDS is sampled by pumping through 6.5m of tubing. I was of the general impression that the pumping of a gas mixture can introduce some "pressure-gradient fractionation". Has it been quantified and could it have a detectable influence on the measurements performed in this study or is the effect too small to affect the results?*

We appreciate this point. In our setup, vapor was sampled through 6.5 m of 1/4" tubing (OD 6.35 mm, ID 5.8 mm) at ~40 mL min$^{-1}$ (instrument pump flow rate). Using site conditions (median T = −5.9 °C, p = 998 hPa), air properties from the ideal-gas and Sutherland relations give $\rho \approx 1.30$ kg m$^{-3}$ and $\mu \approx 1.69\times10^{-5}$ Pa·s.

Flow regime (Reynolds number).

$Re = (\rho\ v\ D) / \mu = (4\ \rho\ Q) / (\pi\ \mu\ D)$

With $Q = 6.667 \times 10^{-7}\ m^3\ s^{-1}$ and $D = 5.8 \times 10^{-3}\ m$, we obtain $Re \approx 11$ (laminar).

Pressure drop (Hagen–Poiseuille law).

$\Delta P = (128\ \mu\ L\ Q) / (\pi\ D^4)$

With $L = 6.5\ m$, this yields $\Delta P \approx 2.6\ Pa$ across the full line.

This pressure drop is negligible relative to atmospheric pressure and far too small to induce measurable pressure-gradient isotope fractionation. Hence, any pumping-related fractionation is negligible compared with the isotopic variability reported in this study.

*Characterization of the soil: Has the ice/vapor in the underlying soil been isotopically characterized? Or at least, do the measurements in the bottom part of the snowpack give a consistent picture of the isotopic nature of the soil vapor (enriched δ 18O and low d-excess if I follow L613)? This seems quite important to me in order to be able to discuss the effects of water vapor flux from entering the snowpack from the ground.*

We did not directly sample the isotopic composition of the underlying soil (neither soil pore vapor nor soil ice) in this study. However, the basal snow and co-located pore-space vapor (0–5 cm) consistently exhibit enriched $\delta^{18}O$ and depressed d-excess relative to overlying layers, particularly during the early/mid periods. This vertical pattern and its persistence when upward temperature gradients favor vapor export from the ground aligns with the reviewer's interpretation of a soil-vapor influence on the bottom of the snowpack. We have clarified this inference in the Discussion near L613, noting that while the signal is consistent with a soil endmember, quantification requires direct soil vapor measurements, which we identify as a priority for future work.

*Access to data: I did not manage to access the restricted data. If I understand correctly, they contain all relevant processed data discussed in the text (meteorological data, snowpack samples data, vapor composition data, precipitation/surface composition data, and snow temperature/humidity data). I trust the authors to have included all that (and maybe some more that I'm forgetting), but I just wanted to be sure.*

Thank you for flagging this. The dataset has been uploaded to our repository and is currently under restricted (embargoed) access for peer review. In line with our policy, the archive will be made public upon publication (DOI listed in the Data Availability statement). If the reviewer would like to inspect the data during review, we will be happy to provide temporary private access upon request.

The repository contains all processed data used in the manuscript.

**Specific Comments**

*L46 - As far as I understand the impacts of snow decrease on soil temperature cannot be simplified to a consistent acceleration of soil warming. Thinner snow covers also favor the cooling of the permafrost during winter. The net impact of a changing snow cover on the ground can either be a warming or cooling effect, depending on the particular conditions at play (e.g. Lawrence and Slater, 2010).*

We thank the reviewer for this clarification. We agree that the effect of reduced snow cover on soil temperature is not unidirectional. A thinner snowpack can indeed enhance soil cooling during winter by reducing insulation, while in spring and summer it may lead to earlier soil warming due to reduced shading and earlier melt onset. We have revised the sentence at L46 to better reflect this dual effect

and now cite Lawrence and Slater (2010) to acknowledge that the net impact depends on seasonal timing and local conditions.

*L65 - I would not use "predominantly" here, as I do not understand what could be a third type of process that is neither a surface nor a sub-surface process.*

We appreciate the reviewer's observation. Our intention was simply to distinguish between surface-driven processes (e.g., sublimation/condensation at the snow–atmosphere interface) and subsurface processes (e.g., vapor diffusion and recrystallization within the snowpack). The word "predominantly" is unnecessary and potentially confusing, so we have removed it in the revised manuscript.

*L95 and elsewhere - For me snow is the resulting (macroscopic) mixture of ice + humid air. Therefore, the notion of (dis)-equilibrium between the air and the snow sounds a bit odd. I understand that sometimes "snow" is used interchangeably with "ice", but since this paper focuses on the distinguishing the isotopic composition of the air phase to that of the ice, it might be worth to use the word "ice" rather than "snow" when specifically discussing the composition of the ice matrix.*

We thank the reviewer for this valuable suggestion. We agree that using the term "snow" to describe the isotopic composition of the solid phase can be misleading, since snow is indeed a mixture of ice grains and pore-space vapor. To avoid ambiguity, we have revised the text throughout to use "ice" when specifically referring to the isotopic composition of the solid matrix and reserved "snow" for the snowpack as a whole (ice + vapor).

*L171 - I do not understand how the calibration and correction of the measurements was performed. The text implies that there is only one a and b per isotopes (so one {a,b} for DHO and another {a,b} for H2 18O), but I do not see how it could be used to get from a raw δX to a corrected δXcorr. Perhaps there is a missing δX in the formula.*

We appreciate the reviewer's careful reading. We have clarified the notation and equations in the Methods. Humidity corrections were applied using a nonlinear regression of the form $\delta X_{corr} = a + b/q$, where $\delta X_{corr}$ is the difference between observed isotopic value and the actual standard isotopic value ($\delta^{18}O$ or $\delta^2H$), $q$ is the water vapor mixing ratio, and $a$ and $b$ are empirically determined constants.

*L173 - Why isn't the correction applied to snowpack vapor measurements as well?*

This humidity correction was applied to both the ambient-air vapor dataset and the snowpack vapor dataset. We have corrected this point in the revised manuscript to avoid confusion.

*L180 - What are the two corrections? There is one for humidity but it is not clear to me what is the other one.*

We thank the reviewer for noting this lack of clarity. In addition to the humidity correction described above, we also applied a scale correction to place the CRDS measurements on the VSMOW2 scale. This was done by measuring liquid standards of known isotopic composition that were vaporized and introduced to the analyser and deriving linear offsets for $\delta^{18}O$ and $\delta^2H$. Together, the two corrections are: (i) the humidity-isotope correction, and (ii) the VSMOW2 scale normalization.

*L231 - I just wanted to mention that the fact that the relative humidity of the vapor was measured within the snow during the whole season and consistently shows saturation is very valuable. It is a crucial piece of information for vapor physics dynamics.*

Thank you for highlighting this point. We have removed the statement implying persistent saturation. As Reviewer 2 also noted, our in-snow RH probes were unheated (to avoid perturbing the snowpack) and likely experienced frosting, which can artifactually report RH ≈ 100%. Using the CRDS water vapor mixing ratios together with co-located snow temperatures, we infer that pore-space vapor was undersaturated. The Methods now state that in-snow RH probe data are not used quantitatively and that RH over ice is instead computed from CRDS humidity and temperature. We also note in the revised discussion (in response to Reviewer 2's major comment) that processes such as barometric pumping can contribute to pore-space undersaturation, further supporting the interpretation that the snowpack was not persistently saturated.

*L235 - If possible please include a brief description of the snowpack stratigraphy. It doesn't have to be super precise I think, but the potential presence (or absence) of slabs or crusts that impede water vapor movement could be important information.*

We have added a brief, qualitative stratigraphy description. While we did not conduct an extensive grain-crystal analysis, field notes and hand-lens observations indicate: a coarse, low-cohesion basal layer; a distinct mid-pack refrozen ice layer that formed after the first warm event; and finer, more cohesive layers above. We acknowledge that a detailed stratigraphy analysis will be very beneficial and we prioritize this in future work.

*L280 - Perhaps rename the section to precise that it focuses on variations at the seasonal time scale, when the previous section focused on diurnal variations.*

Thank you for the suggestion. We have revised the heading to clarify the time scale:
"3.3 Seasonal Variations in Isotopic Composition of Water Vapor in the Snowpack–Atmosphere Continuum."

*L295 - I understand the strengthening of the correlation between the 5 and 15cm levels could suggest some sort of enhanced mixing in the later period (I'm thinking of wind pumping based on the latter part of the article). But isn't the argument partly countered by the fact that the correlation between the ambient air and the snowpack diminishes over time, that could suggest reduced mixing between the snowpack and the ambient air?*

Rather, late-season conditions, low RH, higher winds, and stronger radiation, introduce additional processes in the ambient air (e.g., enhanced surface sublimation and kinetic fractionation, boundary-layer regime shifts, and episodic advection) that increase high-frequency isotopic variability in the ambient air, which do not readily transmit into the snowpack.

*L479 - Could you explain and/or specify why the relation between ambient δ 18O and T in the early period indicates near-equilibrium? The equilibrium fractionation factor decreases with temperature. I would thus expect a negative correlation between δ 18O and T if the ambient vapor was in isotopic equilibrium with some ice source. But perhaps the authors meant something else.*

Thank you for this valuable comment. We agree that if the ambient vapor were in strict isotopic equilibrium with local ice, a negative correlation between $\delta^{18}O$ and temperature would be expected because the equilibrium fractionation factor increases with decreasing temperature. In our case, the observed positive $\delta^{18}O$-T correlation in early winter instead reflects the hydrological cycle isotope–temperature relationship, where colder air masses are isotopically more depleted due to upstream condensation, and warmer air masses are less depleted. This suggests that during the early period, boundary-layer stability limited snowpack–atmosphere exchange, allowing the vapor isotopes to

closely follow regional air mass history and radiative cooling, rather than snow surface equilibrium. We have revised the text accordingly for clarity.

*L618 - I do not follow why temperature serves as a diagnostic of the diffusion-advection transition. Could you elaborate?*

We thank the reviewer for this comment and agree that our original phrasing could be misleading. We have rephrased the text accordingly. Barometric pumping was active throughout the season (new analysis, see response to major comment of Reviewer 2), but periods of increasing temperature coincided with low relative humidity and high wind speeds, which together caused ambient vapor d-excess to rise. These enhanced d-excess signals were then pumped into the snowpack, reinforcing advective transport. Thus, temperature itself is not the mechanistic driver of the diffusion–advection transition, but it acts as a covariate indicating boundary-layer conditions (low RH, high wind, elevated vapor d-excess) that favoured ventilation over diffusion.

**Technical Comments**

*L123 - What is the depth of the ground sensor?*

I assume to you mean the soil temperature sensor. I was at a depth of 5 cm. This has been added to the text

*Fig 1. - If possible, it might be interesting to put a picture of the actual setup. It could be done in the supplementary material not to clutter the main part of the article.*

[Figure]

[Figure]

[Figure]

**Figure 1.** Field and laboratory setup of the vapor isotope monitoring system. (a) Deployed tower at the University of Alaska Anchorage showing vapor intakes at 5 cm and 15 cm above ground within the snowpack and the 1.5 m ambient air intake, along with meteorological sensors measuring RH, T, wind speed, and pressure. Heated and insulated tubing connected each inlet to the isotope analyzer. (b) Laboratory assembly and testing of the system prior to field installation. (c) Field deployment after snow accumulation.

*L151 - Is there a reason why the fourth line was not included in the study in the end?*

he original purpose of the fourth inlet was to measure vapor isotopic composition just above the snowpack surface (<5 cm). However, as the season progressed this inlet was repeatedly buried by new snowfall, requiring frequent re-adjustments to keep it near the surface. This made the vertical position of the inlet inconsistent over time, complicating comparisons with the fixed-height inlets (5 cm, 15 cm,

1.5 m). In addition, snow captured around the inlet likely altered the vapor in its immediate vicinity, introducing potential artifacts. To ensure comparability and consistency across measurement levels, we decided not to include data from this fourth inlet in the analysis. We now clarify this in the revised Methods section.

*L171 - Just out of curiosity, is there a theoretical reason to search the correction under this specific form, or it is based on the shape of the curves in Fig. S1?*

The correction function we used does not arise from a strict theoretical model of isotope fractionation with humidity. Instead, the form was chosen empirically to best represent the calibration data (Fig. S1), consistent with the nonlinear humidity dependence reported in earlier CRDS studies (Weng et al. 2020). In other words, the "shape" of the function is based on the data rather than a first-principles derivation, but it is physically motivated in that humidity-dependent optical effects in the cavity are known to produce such nonlinear behaviour.

*L171 - The function yields δXcorr but only δX is presented instead in the text.*

$\delta X_{corr}$ is the difference between observed isotopic value and the actual standard isotopic value this has been corrected in the manuscript

*L179 - It seems that this sentence just restates what has been said in the paragraph above (standards and ambient measurements are corrected with some non-linear function of q).*

This sentence has been removed as it was redundant

*L223 and Fig. 2 - Precise that this is the relative humidity of the overlying atmosphere (not to be confused with the relative humidity of the snow vapor)*

Thanks this has been corrected to specify ambient relative humidity both in the text and the figure caption

*L255 and L629 - I think the word "more" is missing.*

Thanks this has been added

*L380 - From what I understand, $\Delta\delta^{18}O = \delta^{18}O_{15cm} - \delta^{18}O_{5cm}$ (the difference in isotopic composition between spatial points in the snowpack, which can be applied both for the ice matrix and the vapor). If so, please indicate it clearly so it is not confused with the ice-vapor isotopic difference.*

Thanks, the sentence has been rephrased

*L379 to 396 - The split between the two paragraphs is a bit strange to me. It is §{Δδ 18O in ice} then §{Δδ 18O in vapor + Δd-excess in ice and vapor}. I would rather expect §{Δδ 18O in ice and in vapor} and then §{Δd-excess in ice and vapor}.*

The paragraphs have been restructured

*L497 - I think there are Δ missing: isn't it the Δδ 18O and Δd-excess that respond to ΔT?*

Thanks, Δ has been added

*L535 - It seems that the last part of the paragraph is a just a re-wording of what has been stated above (namely that surface sublimation release high d-excess vapor, that then increase the d-excess of the ambient air and of the deeper snow when wind pumping is present). Consider removing it to lighten the text.*

Thanks, the redundant text has been removed

*L596 - Δδ 18O is already defined as the difference in δ 18O between layers. Perhaps simply use the name " 18O disequilibrium (defined as measured δ 18O minus the theoretical δ 18O equilibrium value)".*

Thanks this has been changed

*Supplementary Material L3 - Remove "com".*

Done
* * *
References:

Weng, Y., Touzeau, A., and Sodemann, H.: Correcting the impact of the isotope composition on the mixing ratio dependency of water vapour isotope measurements with cavity ring-down spectrometers, Atmos. Meas. Tech., 13, 3167–3190, https://doi.org/10.5194/amt-13-3167-2020, 2020.

Lawrence, D. M., & Slater, A. G. (2010). The contribution of snow condition trends to future ground climate. Climate Dynamics, 34(7), 969–981. https://doi.org/10.1007/s00382-009-0537-4

---

## Author Comment (AC3)

**Response to Reviewer #3 for EGU-2025-2724:** *Isotopic Stratification and Non-Equilibrium Processes in a Sub-Arctic Snowpack*

We sincerely thank Reviewer 3 for their careful and detailed evaluation of our manuscript. We recognize the importance of critically examining potential artifacts in pore-space vapor measurements, particularly in porous snowpacks where the act of sampling itself can influence the system. The reviewer has raised thoughtful concerns regarding possible pump-induced entrainment of atmospheric air, and we greatly value the opportunity to clarify our experimental design, inlet hydraulics, and the physical basis for our interpretations.

Reviewer 3 highlights an important concern about whether our continuous sampling could disturb the snowpack vapor environment and influence the measured isotopic signals. They argue that the extraction rates might artificially ventilate the snowpack, potentially explaining observed humidity and isotope variability. They note that diurnal humidity variations observed at 5 and 15 cm depths, without corresponding temperature variations, may indicate pump-induced airflow rather than natural processes.

In our response, we provide additional details on the inlet hardware and hydraulics, demonstrating that the frit resistance confines any perturbation to very local scales. We also show that the temporal and vertical structures of our data are inconsistent with steady sampling artifacts. The frits confine any sampling perturbations to millimeter–centimeter scales, as supported by calculated pressure budgets. Consequently, the large-scale replacement effects suggested by the reviewer do not apply to our setup. To clarify this for readers, we will revise the Methods to describe the inlet hardware in detail and include a summary of the supporting hydraulic calculations.

For clarity, we have structured our replies such that the reviewer comment is presented first (in *italics*), followed immediately by our detailed response (shown in blue).

*Review 3:*

*The manuscript argues that difusion and wind ventilation cause non-equilibrium fractionation, which reshapes the water isotopic composition within the snowpack in both the vapor and the snow on hourly to seasonal timescales.*

*The argument is based on a comprehensive dataset of water vapor isotope observations from both above and within the snowpack, combined with snowpack isotope profiles and direct temperature measurements.*

*The authors support their conclusions based on the finding that their measurements of the water vapor isotopic composition show that the pore-space vapor is rarely in isotopic equilibrium with the surrounding ice.*

*Their experimental setup consists of two inlets, located respectively 5 and 15 cm above the ground, which are buried by snow deposited throughout the season. At the beginning of the campaign, the 15 cm inlet is 45 cm below the snow surface, while at the end of the campaign, it is 65 cm below the snow surface.*

*The target question of the manuscript is crucial for understanding the physical processes that affect the climate signal recorded in the stable isotopic composition stored in the ice crystals that comprise the snowpack. Since the early work of Waddington et al., who hypothesized that wind pumping is important for driving the isotopic composition in the snow, discussions and attempts have been made to quantify the vapor transport within the snowpack through direct measurements.*

Unfortunately, it is this reviewer's view that the authors make similar mistakes as previous attempts to measure interstitial water vapor, in that they disregard the influence of their measurements on the medium they are trying to measure, i.e., the interstitial water vapor.

Contrary to the statement on line 640, "However, the small volume of vapor extracted relative to total pore-space vapor produced no systematic artifacts," I will argue below that the volume of vapor extracted in fact is producing artifacts, which prevent the authors from reaching a robust conclusion.

The fundamental problem with their setup is that the authors remove air from the snowpack continuously:

L 149 "The three inactive sampling lines were continuously flushed by a single pump at a total flow rate of 100 ml/min, yielding approximately 33 ml/min through each inactive line"

L152 "The active line connected to the analyzer was sampled at the instrument´s internal flow rate of approximately 40 ml/min.

This means that a constant flow through each inlet line buried in the snow is 33-40 ml/min. As the air for the inlet lines in the snowpack cannot come from the ground, it must come from the atmosphere above the snowpack. This means that a total of 66-80 sccm of air will flow to the combined inlets in the snow. See Figure 1 below as a sketch of the setup. Figure 1: Sketch of setup

[Figure]

Figure 1: Sketch of setup

The fundamental question that then arises is, where does the air come from? It must be such that the air will follow a path of the smallest integrated resistance. This means that the air cannot come from infinity. On the other side, it also seems unlikely that the air follows a tube flow straight from the surface through a path with a diameter of 6 mm equal to the inlet ID.

Below I therefore consider two situations: Conservative option 1: The air enters the snowpack through an area with a diameter of twice the depth of the inlet (90 cm beginning of season and 130 cm end of season) and travels through a semi-sphere of the snowpack. See figure 2 for a sketch of the setup. Figure 2: Sketch of a conservative thought experiment of how the air travels through the snowpack.

[Figure]

Figure 2: Sketch of a conservative thought experiment of how the air travels through the snowpack.

Making the following assumption that the density of the snow is 500 kg/m3 based on the information in the text that the authors observed compaction during periods of no precipitation and the relatively high temperatures at the field site.

For the calculations of snow depth at 60 cm:

The volume of the semi-sphere is 190e3 cm3 . This means that the volume of air is 95e3 cm3 . With a flow rate of respectively 33 or 40 sccm per inlet, this would mean that all the interstitial air will be replaced every 20 to 24 hours.

Even for this relatively conservative estimate, I will therefore argue that the author's argument on line 640, "However, the small volume of vapor extracted relative to total porespace vapor produced no systematic artifacts," does not hold.

A more realistic flow field through the snowpack would probably be better described by a column flow with a diameter of 40 cm from the top of the snowpack down to the inlet. Figure 3 illustrates perhaps a more realistic flow field through the snowpack. Figure 3: Illustration of a perhaps more realistic flow field through snowpack.

[Figure]

*Figure 3: Illustration of a perhaps more realistic flow field through snowpack.*

*Again, carrying out the calculations for a snow depth of 60 cm.*

*In this case, the volume of the column is 56e3 cm3 and the total volume of the interstitial air is 28 cm3 .For the given flow rates through the inlet lines, this would mean that the air in the snow column is replaced every 6 to 7 hours.*

*Based on these calculations, I believe that the effect on the interstitial vapor isotopic composition, which the authors attribute to diffusion and wind ventilation, is a result of the forced transport of atmospheric air through the snowpack.*

*Further support for my conclusion is provided by the authors in the observed snowpack temperature at the depth of the inlet and the observed diurnal variations in humidity of the interstitial vapor:*

*Figure S2 shows that no diurnal variation in snowpack temperature is observed at the 5 and 15 cm inlet*

[Figure]

Figure S2 Plots depicting diurnal patterns in ambient air temperature (C), relative humidity w.r.t. ice (%), wind speed (m/s), and soil and snow temperatures (5cm and 15cm) depth, measured over three distinct periods: early, mid, and late.

However, the authors also demonstrate clear diurnal variations in the observed humidity at the inlets, as shown in Figure 3.

[Figure]

Figure 3 Plots depicting diurnal variations in water vapor concentration (1) and isotopic compositions ($\delta^{18}O$ (2), $\delta^2H$ (3), d-excess (4)) in ambient air (1.5 m) (a) and at 5cm (c), 15 cm (b) within a snowpack and across three distinct periods: blue-early, yellow-mid and magenta-late.

Following the findings of Neumann et al. 2009 (Sublimation rate and the mass-transfer coefficient for snow sublimation): "Our data (e.g. Fig. 4) suggest that the snow sublimates rapidly, and that for snow samples with thickness 1 cm or greater, pore spaces in snow are typically saturated with vapor, as other investigators have assumed [5]"

This means that the snow at the 5 and 15 cm inlet must have a diurnal variation in temperature in order to create a diurnal variation in humidity. As the temperature at 5 and 15 cm does not show any diurnal temperature variation, this means that the flow of air into the inlets must influence the temperature in the vicinity of the inlets, but not be recorded by the temperature observations. Hence, one might conclude that even the "perhaps more realistic flow field in the snowpack" is in fact still too conservative, and the flow from the surface to the inlet follows an even narrower corridor through the snowpack.

Based on my above argument, I therefore believe that the setup in the manuscript of Dar et al. does not allow the authors to reach the presented conclusions

**Response to Reviewer:**

All of the reviewer's points converge on a central issue: whether our sampling significantly perturbs the snowpack vapor environment. In the response below, we provide additional details on inlet geometry, quantify pressure drops across the frits versus the snowpack, and show that any perturbation is confined to millimeter–centimeter scales rather than the decimeter–meter scales suggested by the reviewer's thought experiments. In practice, nearly all of the pressure drop occurs across the frit, thus, the snowpack does not experience the column-scale suction envisioned by the reviewer, and the induced flow field is localized to centimeters around the frit.

Our original manuscript did not describe the buried inlet hardware, which is essential to the hydraulics. Each snowpack intake terminated in a downward-facing, sintered stainless-steel frit (2.54 mm face, 10 µm pore size). This geometry (i) truncates the accessible solid angle (no upward-facing draw from the snow surface) and (ii) introduces a large internal resistance.

[Figure]

*Figure 1 Photograph showing the orientation of the sampling inlets (downward-facing frits at 5 cm and 15 cm above ground) together with the co-located RH/temperature sensor inside its housing. The downward-facing geometry prevents direct line-of-sight entrainment from the snow surface and minimizes artifacts from dynamic pressure fluctuations.*

**1) Hardware makes broad entrainment impossible.**

The reviewer's sketches treat each line's measured flow ($\approx$33–40 mL min$^{-1}$) as if it were supplied from a large region extending to the snow surface, either a hemisphere with radius equal to inlet depth ($\approx$0.45–0.65 m) or a 0.40 m-diameter, 0.60 m-tall column, so that a large pore volume is "replaced" every ~6–24 h. That construction implicitly requires most of the pump head (the suction applied by the analyzer) to be available across the bulk snowpack.

With a downward-facing frit, almost all the pump head is consumed inside the frit. The measured per-inlet flow is 40 mL min$^{-1}$; for a 2.54 mm diameter, "10 µm" frit this requires on the order of $10^2$–$10^3$ Pa across the frit (Kozeny–Carman bound) (Appendix A2). By contrast, when we bound the pressure drop available in the snow using a conservative local control volume (lateral supply from the immediate neighborhood under the frit) and take the inlet heights above ground (5 cm and 15 cm) as upper limits for the local path length, the resulting snow pressure drops are sub-Pa to a few Pa (Appendix A3) for realistic seasonal/fresh-snow permeabilities. Under such small residual drops, Darcy scaling confines the radius of the perturbed zone to millimeters–centimeters, not decimeters (Appendix A4).

Because the frit dissipates essentially all the pump head, the snow is not hydraulically driven as a half-meter hemisphere or a 40-cm-wide column. The draw is local (mm–cm scale), and the reviewer's replacement-time arithmetic for large volumes does not diagnose our downward-facing frit configuration.

We will add (i) a concise description of the inlet geometry and (ii) a reference to the calculations below, which document the head budget, the local snow pressure drops, and the resulting perturbed-zone size.

**2) The flow regime is slow, laminar, and well within Darcy's law (Appendix A5)**
Both inside the frit and in the snow, Reynolds numbers are much less than one. This ensures creeping, linear Darcy flow. There is no possibility of jetting or channel-like flows that might connect the inlet directly to the ambient air.

**3) The temporal structure of the data contradicts an artifact.**
If sampling artifacts were important, they would produce a slow, quasi-steady drift of pore vapor toward ambient conditions over the 6–24-hour turnover times implied by the reviewer's volumes. Instead, our measurements show sharp, episodic excursions that align with external forcing. We observe persistent differences between 5 cm and 15 cm vapor that evolve through the season in systematic ways (Fig 4 in the manuscript). These vertical gradients scale linearly with measured temperature differences, exactly as expected for gradient-driven diffusion, and cannot be explained by external suction. We perceive transitions are evidence of physical vapor transport processes, not constant biases from sampling.

**Response to "no T cycle at 5–15 cm means RH cycles are pump artifacts"**

- **Neumann et al. (2009) does not directly apply to our field setting.** Neumann's lab samples (1–5 cm thick) show the air exiting a thin sample is typically saturated, with saturation reached within roughly the first millimeter to centimeter of snow; they also note that undersaturation can occur in large-pore layers or at higher flows. This does not imply that RH at 5–15 cm depth in a field snowpack can only vary if local temperature varies.

- **Why RH can vary at depth without a local temperature cycle.** RH equals $e/e_s(T)$. Even if temperature at 5–15 cm is flat (so $e_s$ is steady there), diurnal swings in ambient temperature, ambient RH, and wind change the surface boundary: colder ambient air lowers $e_s$ at the surface (Clausius–Clapeyron), and wind-driven pressure gradients periodically ventilate the near-surface snow (wind pumping). These forcings create vertical gradients in vapor pressure ($e$) that drive time-varying diffusive and advective fluxes into the pack. As a result, $e(z,t)$ and therefore RH can oscillate at 5–15 cm without any measurable local temperature cycle and without a pump-forced corridor.

- **New barometric pumping evidence.** Our time-shift analysis demonstrates that snowpack vapor responds to barometric pressure oscillations on sub-hourly to hourly scales (details in response to major comment of Reviewer 2). These timescales are much shorter than molecular diffusion (3–10 h at our depths) and show that pressure-driven advection is active throughout the record. This barometric pumping efficiently propagates humidity and isotopic variability into the snowpack, explaining the observed RH cycles at 5–15 cm without requiring a local temperature cycle.

- **Laboratory evidence under isothermal conditions.** Ebner et al 2017. forced saturated airflow through snow under isothermal and temperature-gradient conditions and observed nonequilibrium vapor–snow interaction and large $\delta^{18}O$ changes. This demonstrates that humidity and isotope signals propagate in snow without co-located temperature cycles at depth.

The claim that RH cycles at 5–15 cm require a local temperature cycle or a narrow pump corridor is unsupported. Surface-forced ventilation is sufficient to explain the observed RH cycles at depth.

To summarize, the hydraulics of the frit–snow system, supported by our calculations and external literature, show that pump-induced artifacts are limited to the immediate vicinity of the inlets and cannot explain the observed vertical isotope gradients, their temporal variability, or seasonal transitions in isotopic composition. We therefore maintain that the observed signals reflect physical processes of diffusion and ventilation within the snowpack.

**APPENDIX:**

A1. Constants and measured quantities

- Volumetric flow (instrumental flow):
  $Q=40$ mL/min$=(40\times10^{-6})/60=6.67\times10{-7}$ m$^3$/s$= 6.67\times10^{-7}$ m$^3$/s.

- Air viscosity: $\mu=1.7\times10{-5}$ Pa s

- Air density (for Reynolds): rho $\approx 1.3$ kg/m$^3$

- Frit face: diameter 2.54 cm, radius rf$=12.7$ mm area Af$=\pi rf^2=5.067\times10^{-4}$ m$^2$

- Superficial velocity through frit: qf$=Q/Af=1.316\times10^{-3}$ m/s

A2. Frit pressure drop

Permeability of the frit: Nominal "10 µm" frit, use pore size d_p $= 1.0 \times 10^{-5}$ m as an engineering proxy. Kozeny–Carman to calculate the frit permeability:
$k\_f = d\_p^2 \times eps^3 / [180 \times (1 - eps)^2]$

We bracket porosity eps $= 0.25$–$0.45$ and thickness L_f $= 1$–$3$ mm

Darcy's law, the fundamental relation for fluid flow through porous media is used to describe the pressure drop when air passes through the frit of thickness L_f

$\Delta P\_frit = \mu * (L\_f / k\_f) * q\_f$

*Results:*

[Figure]

*Figure 2 Pressure drop across the frit (ΔP_frit) as a function of (a) porosity at fixed thickness Lf=2 mm, (b) frit thickness at fixed porosity ε=0.35, and (c) permeability at fixed thickness Lf=2 mm.Porosity sweep at L_f = 2 mm:*

**Hence, ΔP_frit is robustly $10^2$ to$10^3$ Pa at the measured flow.**

A3. Snow pressure drop for a local lateral supply:

The frit faces downward; the intake is lateral from a local neighborhood. We model a cylinder of radius $r_c$ and length $L_s$ that supplies the frit. Because the frits are 5 cm and 15 cm above ground, we use $L_s \le 5$ cm (lower inlet) and $L_s \le 15$ cm (upper inlet) as upper bounds. We take two footprints $r_c = 5$ cm and 10 cm. The frit face radius is ~1.27 cm. Choosing rc=5 cm already represents a local neighborhood that extends ~4× the frit radius in all directions, well beyond the hardware footprint and into the surrounding snow. It is a reasonable lower bracket for a local supply region. rc=10 cm is intentionally large: its cross-sectional area is ~62× the frit face area. Taking such a large rc demonstrates conservatism

Snow permeability sweep: $k = 10^{-11}$–$10^{-7}$ m$^2$ (Calone et al. 2012)

Darcy's law across snow:
$\Delta P\_snow = mu * (L_s / k) * (Q / (pi\ r_c^2))$

One worked example:
$k = 1e{-}10$ m$^2$, $L_s = 0.05$ m, $r_c = 0.05$ m.
$Q/(pi\ r_c^2) = 6.67e{-}7 / [pi*(0.05)^2] = 8.49e{-}5$ m/s.
$mu*( L_s /k) = 1.7e{-}5 * (0.05 / 1e{-}10) = 8.5e3$ Pa.
$\Delta P\_snow = 8.5e3 * 8.49e{-}5 \approx 0.72$ Pa.

Results:

[Figure]

*Figure 3 Pressure drop across the snow layer (ΔP_snow) as a function of snow permeability (k) for two cylinder radii (rc = 5 cm and rc = 10 cm) and two snow thicknesses (Ls = 5 cm and Ls = 15 cm). Results are shown on log–log axes*

**This plot shows that even at the lowest permeabilities ($10^{-11}$ m$^2$), the snow pressure drop remains far smaller than the frit drop ($10^2$–$10^3$ Pa), so the frit overwhelmingly dominates the hydraulic resistance.**

A4. Radius of the perturbed zone

When flow is drawn through it, the frit creates a small pressure disturbance in the snow directly beneath. We want to estimate how far sideways that disturbance spreads, the lateral reach, called a.

Radial Darcy inflow (hemispherical assumption)

If we imagine flow lines spreading out like a hemisphere beneath the frit, Darcy's law gives:

$$Q = 2\pi k \, (\Delta P_{snow}/\mu) \, a \qquad\qquad (1)$$

Here $Q$ = flow rate through the frit, $k$ = permeability of the snow, $\mu$ = air viscosity, $\Delta P_{snow}$ pressure drop in the snow, $a$ = perturbed radius

Control-volume Darcy relation:

We can also think of the snow under the frit as a cylindrical plug of radius $r_c$ and depth $L_s$. In that case:

$$\Delta P_{snow} = \mu \, ( \, L_s / k) \, (Q/\pi r_c^2)$$

$$k \, \Delta P_{snow}/\mu = L_s \, (Q/\pi r_c^2) \qquad\qquad (2)$$

Where $r_c$ = frit radius, $L_s$ = penetration depth of the flow beneath the frit

Now substitute (2) into (1):

$$Q = 2\pi [L_s \, (Q/\pi r_c^2] a$$

$$a = r_c^2/2L_s$$

So the disturbed radius depends only on geometry: frit radius and penetration depth.

*Example Calculations:*

- $r_c$ =5 cm, $L_s$ =5 cm ; a=2.5cm

- $r_c$ =5 cm, $L_s$ =15 cm ; a=0.83cm

- $r_c$ =10 cm, $L_s$ = 5 cm ; a=10 cm

- $r_c$ =10 cm, $L_s$ =5 cm ; a=3.3 cm

**Hence, the perturbed zone radius is only millimeters to a few centimeters, not tens of centimeters.**

A5. Flow regime (including porosity and tortuosity ranges)

We evaluate the flow regime in both the frit and the surrounding snowpack using interstitial velocities, effective pore dimensions, and Reynolds numbers.

- Frit interstitial velocity: $u_i = q_f/\varepsilon$, where $q_f$ is the superficial velocity through the frit and $\varepsilon$ is the frit porosity.

- Snow interstitial velocity: $u_i = u/\phi$, $u = Q/(\pi r_c^2)$, Q is the volumetric flow rate, $r_c$ is the intake radius, and is the snow porosity.

- Effective pore length: $d_{eff} = d_p/\tau$, where $d_p$ is the mean pore diameter and $\tau$ is the tortuosity.

- Reynolds number: $R_e = \rho u_i d_{eff}/\mu$, where $\rho$ is the air density and $\mu$ is the viscosity.

Frit:

- With pore sizes of about $1\times10^{-5}$ m, porosity between 0.30–0.40, and tortuosity 1.1–1.4, the superficial velocity through the frit is $1.3\times10^{-3}$ m/s (baseline flow)

- Interstitial velocities range from about 3.3 to $4.4\times10^{-3}$ m/s.

- Effective pore sizes fall in the range 7–9 μm.

- The corresponding Reynolds numbers are very low: 1.8–3.1×10$^{-3}$.

Snow:

- For snow, we allowed pore sizes between 1×10$^{-4}$ and 1×10$^{-3}$ m to represent the variability from dense wind slabs to fresh snow. Porosity was taken between 0.55–0.80, and tortuosity between 1.1–1.4.

- At rc=5r_c = 5rc=5 cm:

    - Superficial velocities are 8.5×10–m/s (baseline)

    - Interstitial velocities span 1.1–1.5×10–4 m/s

    - Effective pore sizes range from 0.07 to 0.9 mm.

    - Reynolds numbers are 5.8×10–4–1.1×10–2

- At rc=10r_c = 10rc=10 cm:

    - Superficial velocities are 2.1×10–5 m/s

    - Interstitial velocities span 2.7–3.9×10–5 m/s.

    - Effective pore sizes range from 0.07 to 0.9 mm.

    - Reynolds numbers are 1.5×10–4–2.7×10–3

Creeping (Darcy/Stokes) flow holds; no inertial or turbulent enhancement is available to drive bulk flushing.

For the conservative high-flow case (Q = 73 mL/min = 40 + 33):

- Frit still dominates the pressure budget. Scaling ∝ Q, ΔPfrit rises to roughly 2×10$^{2}$ to 2×10$^{3}$ Pa, remaining 1–3 orders larger than any snow drop over local paths.

- Snow pressure drops remain small over local supply lengths.
  For k=10−11–10−7 m$^{2}$, r_c = 5–10 cm, Ls=5–15 cm:
  ΔPsnow ~ 3.3×10$^{-4}$ to 4.0×10 Pa .

- Perturbation footprint is unchanged (set by geometry).
  Flow regime remains creeping.
  With φ=0.55–0.80, τ=1.1–1.4

    - rc=5cm → Re ≈ 3.2×10$^{-3}$ to 5.9×10$^{-3}$

    - rc=10 cm → Re ≈ 7.9×10$^{-4}$ to 1.5×10$^{-3}$
      All Re ≪ 1 → Darcy/Stokes; no inertial/turbulent flushing available.

Even at this conservative, higher draw, the frit's hydraulic resistance confines perturbations to the immediate intake neighborhood, and cannot drive bulk snowpack flushing.

**References:**

Ebner, P. P., Steen-Larsen, H. C., Stenni, B., Schneebeli, M., & Steinfeld, A. (2017). Experimental observation of transient $\delta^{18}$O interaction between snow and advective airflow under various temperature-gradient conditions. *The Cryosphere, 11*, 1733–1743. https://doi.org/10.5194/tc-11-1733-2017

Neumann, T. A., Albert, M. R., Engel, C., Courville, Z., & Perron, F. (2009). Sublimation rate and the mass-transfer coefficient for snow sublimation. *International Journal of Heat and Mass Transfer, 52*(1–2), 309–315. https://doi.org/10.1016/j.ijheatmasstransfer.2008.06.003

---

## Author Comment (AC4)

Review of EGU-2025-2724: "Isotopic Stratification and Non-Equilibrium Processes in a Sub-Arctic Snowpack"

We thank the reviewer for the careful reading and constructive comments. Below, reviewer comments are shown in *italics*, and our responses are given in **blue text**.

Major comment:

*I'm a bit concerned about the correctness of the measurement looking at the data in Figure 2 and 3. Your temperature measurement inside the snowpack (Figure 2) shows a minimum of around -4 degrees (corresponding to a saturation vapor concentration of around 4300 ppmv) but your concentration measurements with the Picarro (Figure 3) show a minimum of around 2700 ppmv. This would indicate that you have a relative humidity of only 65% inside the pore space of the snow pack which is not possible. Do you have an explanation for this? The same for the ambient measurement, you measured the lowest temperature at 1.5m of around -20 degrees (corresponding to a saturation vapor concentration of around 1500 ppmv) but your lowest measured concentration was around 2500 ppmv. This would indicate that your ambient air was sometimes oversaturated. Or could it be that your Picarro sucked in ice/water particles or is the Campbell Scientific AP200 intake also used for the isotope vapor? Maybe you could provide a picture of the system in the supplement materials.*

*You claimed that the pore spaces of the snowpack was at saturation but I could not find any evidence in the data. Therefore, it seems like that either your setup was not leak-tight or there is an offset in the humidity measurement of your Picarro.*

**Response to major comment:**

Our response is structured in three steps: first confirming that the system was leak-tight, then validating Picarro humidity against an independent RH sensor for ambient air, and finally explaining why undersaturation in the pore space is physically plausible.

Leak tightness of the sampling system was verified prior to measurements using a **pressure hold test**. Each sampling line was connected to a dry air cylinder, pressurized, and then isolated by closing the cylinder valve. With the line sealed, the internal pressure was monitored for 30 minutes while sequentially selecting each line via the multiplexer. The pressure remained stable throughout these tests, confirming that the sampling lines were leak-proof.

The Campbell Scientific AP200 inlet was indeed used for isotope vapor sampling, but the system was equipped with heated and insulated lines and terminated in downward-facing sintered frits (2.54 mm diameter, 10 μm pore size) to prevent entrainment of ice/water particles. This configuration minimizes the risk of *sucked in ice particles*. The picture of the setup is included now.

We verified the ambient air measurements against a co-located RH sensor at 1.5 m height within a radiation shield. Converting the Picarro L2130-i $H_2O$ concentrations to relative humidity over ice using the Goff–Gratch formulation shows very good agreement with the independent RH sensor ($R^2$ = 0.91, RMSE = 5.3 %RH, mean bias = −2.2 %RH; Fig. 1a). A direct comparison in absolute humidity (ppmv) likewise demonstrates excellent consistency between the CRDS and the sensor ($R^2$ = 0.99, RMSE = 214 ppmv, mean bias = −116 ppmv; Fig. 1b). These validations confirm that there is no significant

offset in the Picarro humidity measurement.

[Figure]

*Figure 1 Comparison of humidity measurements from the Picarro L2130-i CRDS and a co-located independent RH sensor at 1.5 m height. (a) Relative humidity with respect to ice (RHice) derived from CRDS $H_2O$ concentrations versus RHice from the RH sensor, converted from RH over liquid using the Goff–Gratch formulation. (b) Absolute humidity (ppmv) from the CRDS versus the sensor (converted from RHice and ambient temperature/pressure). The red dashed line indicates the 1:1 relationship. Statistics shown are coefficient of determination ($R^2$), root mean square error (RMSE), and mean bias (CRDS minus sensor). Both comparisons demonstrate very good agreement, confirming the absence of a systematic offset in the CRDS humidity measurements.*

You are right to question our earlier statement that the snowpack pore space was saturated. That claim was based on buried RH sensors that were not heated. Since unheated RH sensors in snow are prone to frost deposition and tend to report near-constant values close to 100%, we do not rely on these measurements to assert saturation. We have removed this statement from the manuscript. The apparent undersaturation inside the snowpack is real rather than an artifact: laboratory and modelling studies have shown that deep snow layers can indeed be undersaturated relative to ice due to limited sublimation kinetics (e.g., Ebner et al., 2017; Bouvet et al., 2024). Our pore-space measurements therefore likely captured genuine undersaturation, however the saturation we observed was larger in magnitude than reported in those studies.

Importantly, prior studies of gas transport in snowpacks demonstrate that ventilation by barometric and wind-driven pressure fluctuations is an established process that operates on sub-hourly timescales. For example, Massman (1995) combined field data and modeling to show that turbulent pressure fluctuations penetrating a 2 m snowpack enhanced $CO_2$ fluxes by 19–31% above diffusion alone. In follow-up modeling, Massman (1997) showed that barometric and turbulence-driven oscillations can penetrate meters into the snowpack and drive advective velocities on the order of $10^{-4}$ m s$^{-1}$, with short-term fluxes deviating by up to ±25% from diffusion. Graham and Risk (2018) later confirmed in field observations that $CO_2$ concentrations inside a seasonal snowpack can drop by hundreds of ppm within hours during windy episodes, directly evidencing the timescale of ventilation events. These studies focused on $CO_2$ and other trace gases, which do not equilibrate with ice surfaces, but the physics of air exchange is the same for all pore-space gases. The key implication for our work is that pore air can be replaced by ambient air far faster than molecular diffusion or sublimation can re-establish local vapor equilibrium with ice. For $H_2O$ vapor, this means that

ventilation can continually import drier ambient air, maintaining undersaturation inside the snowpack. Thus, while $CO_2$ studies demonstrate the transport mechanism, our measurements show the water vapor consequence: persistent deviations of pore humidity from ice saturation are a physically plausible outcome of rapid ventilation.

In our original submission we had only considered wind-driven ventilation as the mechanism of pore-space exchange. Following the reviewer's comment and our re-analysis, we now explicitly include barometric pumping of ambient air into the snowpack as an additional process. As a baseline, we first evaluated diffusion timescales (Figure 2). Purely diffusive equilibration of pore-space vapor with the surrounding snow requires 3–12 hours depending on depth and snow conditions. This sets a relatively slow reference against which pressure-driven processes can be compared. By contrast, our new calculations of the pressure forcing required for ventilation (Figure 3) show that barometric pressure variations routinely exceed this threshold at both 5 and 15 cm, often by more than an order of magnitude. Wind-driven forcing alone only intermittently reaches the required ΔP, but when it does, it acts in concert with barometric oscillations to further enhance exchange.

Together, these results demonstrate that ventilation is physically feasible across much of the record, primarily through barometric pumping. This means that pore air was frequently mixed with intruding ambient air on sub-hourly timescales, far faster than diffusive equilibration could occur. Such a separation of timescales explains why persistent undersaturation developed in the pore space: intrusions of drier ambient air occurred more rapidly than sublimation from ice surfaces could replenish vapor toward saturation.

**Diffusion timescales (Figure 2, Calculation procedure in Appendix A1).**
The diffusion timescale provides a baseline reference for the characteristic time it takes water vapor molecules to traverse a given snow depth by molecular diffusion. This is critical because if natural forcings (wind or barometric pressure changes) act on shorter timescales than diffusion, then ventilation can outpace diffusive exchange and drive pore-space conditions away from equilibrium. Conversely, if natural forcings are slower than diffusion, then pore vapor has enough time to equilibrate, and deviations are less likely to persist. At the 5 cm basal intake, diffusion times were consistently longer than at 15 cm, because the effective path length to the surface was greater (L=snow depth−0.05m vs L = snow depth - 0.15m). Early in the season, diffusion times were ~5-6 hours at 5 cm compared to ~3-4 hours at 15 cm. As the snowpack accumulated diffusion times increased reaching ~12 hours at 5 cm and ~9-11 hours at 15 cm. Step increases in the curves coincide with snowfall events that increased snow depth, since t_diff grows with the square of the diffusion path length. The shaded bands reflect the uncertainty in tortuosity ($\tau$ = 1.1–1.4). This parameter affects the absolute magnitude of t_diff but does not change the overall seasonal pattern.

**Why RH_ice < 100% is physically plausible in our record:**

Reviewer raised the concern that our pore-space vapor concentrations imply relative humidities below ice saturation, which would seem contradictory to the expectation of fully saturated snowpack air. Here we explain why such undersaturation is physically plausible in our measurements.

Figure 3 compares the pressure gradients required to ventilate the snowpack pore space with the magnitudes of two natural forcing mechanisms: wind pumping and barometric oscillations. The analysis was conducted for lag times of 15, 30, and 60 minutes, which were chosen to represent short to intermediate exchange timescales relevant to pore–surface ventilation. These windows were derived from the underlying 5-minute observations of wind speed and barometric pressure, and are consistent with the temporal averaging applied to the isotopic dataset, which was aggregated to hourly resolution (see Methods in the original manuscript). By testing multiple lag lengths, the analysis

brackets both transient and sustained forcing events that could influence vapor exchange in the snowpack.

**Required ΔP (left column, Calculation procedure in Appendix A2):**

The shaded brown envelopes represent the range of pressure differences needed to ventilate the snowpack, a range of realistic seasonal snow properties: Permeability: $10^{-11}$ to $10^{-7}$ m$^2$, Porosity ($\phi$): 0.55 to 0.80 (Calonne et al., 2012). Tortuosity ($\tau$): 1.1 to 1.4 (Lieblappen et al., 2020). On the log scale, the required values mostly fall between $10^{-2}$ and $10^1$ Pa. Shorter lag times (15 min) demand higher gradients, reaching up to ~10 Pa ($10^1$ Pa), while longer lag times (60 min) require much less, often close to $10^{-2}$–$10^{-1}$ Pa. Seasonal changes shift the envelopes slightly, but lag time is the dominant control: as the equilibration window lengthens, progressively smaller pressure perturbations are sufficient.

**Wind forcing (middle column, Calculation procedure in Appendix A3):**

Dynamic pressure from wind events produces values that typically fall between $10^{-1}$ and 1 Pa, occasionally spiking toward 10 Pa during strong gusts. Compared to the required range, this means wind forcing can occasionally overlap with the ΔP needed for 15–30 min ventilation windows, but for much of the winter it remains too weak ($\leq 10^{-1}$ Pa). Thus, wind pumping appears as an episodic driver of ventilation, tied to short bursts of strong wind.

**Barometric forcing (right column, Calculation procedure in Appendix A3):**

Oscillations in surface pressure are consistently large on the log scale, ranging from ~1 Pa ($10^0$ Pa) up to >100 Pa ($10^2$ Pa), and in some cases approaching 1000 Pa ($10^3$ Pa). These magnitudes exceed the required ΔP envelopes across all lag times by more than an order of magnitude. Importantly, unlike wind, barometric oscillations are persistent rather than intermittent, providing a sustained mechanism for sub-surface ventilation throughout the season.

When viewed together, the log-scaled comparisons show a strong asymmetry: wind pumping provides sporadic pressure pulses that only sometimes reach the required $10^0$–$10^1$ Pa range, while barometric forcing is continuous and consistently exceeds the required ΔP by one to three orders of magnitude (10–1000 Pa vs. 0.01–10 Pa). However, whether this forcing translates into actual equilibration also depends on the diffusion timescales (Figure 2), which remain on the order of 6–12 hours in late winter. This helps explain why pore-space vapor in our Picarro measurements can remain undersaturated relative to equilibrium, even when external forcings appear more than sufficient.

Figure 4 provides a categorical, time-resolved view of ventilation feasibility under average snowpack conditions. Each vertical stripe indicates whether the available forcing (wind, barometric, or both) exceeded the required ΔP in 15, 30, or 60 min windows. Consistent with the magnitude-based comparison in Fig. 3, barometric oscillations dominate: exceedances are flagged almost continuously across the season (red). Wind forcing (green) rarely exceeds the threshold alone, but occasionally coincides with barometric events (purple). Grey intervals indicate times when neither mechanism was sufficient, which occur mainly at short lag windows (15 min). By 30–60 min windows, barometric oscillations alone are nearly always strong enough to ventilate the pore space. This barcode representation therefore provides a complementary, threshold-focused confirmation of the Fig. 3 result: ventilation is physically feasible primarily through barometric pumping, with wind playing only an episodic role.

[Figure]

Figure 2 Estimated molecular diffusion timescales for pore-space vapor transport in the snowpack. Diffusion time was calculated as the squared transport distance divided by the effective diffusivity, where the latter accounts for a tortuosity range ($\tau$ = 1.1–1.4). Solid blue line = mean diffusion time for 5 cm depth, dashed red line = mean diffusion time for 15 cm depth. Shaded bands represent the range across the tortuosity values. Diffusion times are consistently longer at 5 cm than at 15 cm because of the greater distance to the snow–atmosphere interface. Temporal variability reflects evolving snow depth, with diffusion times spanning from ~3–5 hours in early winter to >10 hours during February–March. When compared to the forcing plots (Fig. 2), these diffusion times indicate that barometric pressure oscillations, which occur at sub-hourly to hourly scales, can easily outpace molecular diffusion and drive ventilation events.

[Figure]

*Figure 3 Comparison of required and observed pressure gradients for pore-space ventilation at 5 cm and 15 cm depth. Each row corresponds to a prescribed lag window of **15 min** (top), **30 min** (middle), and **60 min** (bottom). **Left column (Required ΔP):** Pressure difference required to drive air advection across the snowpack depth within the lag window, calculated from Darcy's law using a broad range of snow properties. Shaded envelopes show the min–max range across property combinations. Solid black line = mean at 5 cm depth; dashed black line = mean at 15 cm depth. Shorter lag times correspond to higher required ΔP values. **Middle column (Wind forcing):** Dynamic pressure forcing estimated from maximum wind speeds in each lag window. Spikes correspond to gust events that drive strong but intermittent pressure loads. **Right column (Barometric forcing):** Pressure oscillations associated with barometric variability, quantified as the peak-to-trough swing within each lag window from atmospheric pressure measurements. All axes are log-scaled for ΔP. This side-by-side framework enables direct comparison of required versus available pressure forcing, illustrating when wind and barometric variability are sufficient to ventilate the snowpack faster than molecular diffusion (Figure 2).*

[Figure]

*Figure 4 Categorical exceedance of required pressure gradients for ventilation under average snowpack conditions. Panels show 15, 30, and 60 min lag windows (columns) for pore-space intakes at 5 cm and 15 cm depth (rows). Colors indicate which forcing mechanism exceeded the required ΔP: barometric only (red), wind only (green), both (purple), or neither (grey). Results confirm that barometric oscillations alone are sufficient to drive ventilation across most of the record, particularly at longer windows, while wind pumping is only intermittently strong enough to contribute.*

*Sometimes you are talking about that vapor measurements were taken at 5 or 15 cm depth. This is a bit misleading because when you are talking about the depth some readers will see it as the distance from the surface into the snowpack. I would suggest to skip the 'depth' or call it 'height' to improve the readiness.*

We thank the reviewer for pointing this out. In our setup, vapor intakes were positioned at fixed heights above the ground, not as distances below the snow surface. We agree that calling them "depths" is potentially misleading. We have revised the manuscript to use "height within the snowpack" (e.g. "5 cm height" and "15 cm height") throughout and clarified in the Methods that these are measured upward from the ground.

*I'm missing important information to get a better understanding of the setup and to interpret the results. How did you check that your system is leak-tight and did you perform a humidity correction of the Picarro?*

Leak-tightness: Prior to field deployment, each sampling line was connected to a dry air cylinder, pressurized, and isolated by closing the cylinder valve. Pressure in the sealed line was monitored for 30 minutes while sequentially switching lines via the multiplexer. No pressure decay was observed, confirming that all lines were leak-proof.

We did not apply a humidity correction to the Picarro data; instead, we validated the raw humidity measurements against a co-located RH sensor. This comparison showed very good agreement ($R^2$ = 0.91, RMSE = 5.3 %RH, mean bias = −2.2 %RH), confirming that no systematic offset was present. Details are provided in our response to the major comment. We will add details on both leak testing and clarified that humidity-isotope correction in the revised Methods.

*I'm a bit concern about your comparison of your results between the three different periods (early, mid and late) because the snow depth is not constant and is changing up to 30 cm during the campaign. I would suggest to provide an explanation why you can still compare the different periods (especially about the data of the two intakes within the snowpack) with each other.*

We agree that snow depth changed substantially during the campaign. However, the intake heights were fixed at 5 cm and 15 cm above ground, such that they remained within the snowpack throughout the entire measurement period, even as total depth increased. The intakes therefore sampled pore vapor air at consistent vertical positions relative to the ground.

While the overlying snow depth did vary, this primarily altered the boundary conditions above the intakes (e.g. thickness of snow cover, permeability to ambient exchange), rather than the absolute meaning of the intake heights. Thus, comparing the three periods provides insight into how changes in snowpack thickness and atmospheric forcing affected the same pore-space levels. We now clarify this in the revised Methods and Discussion and emphasize that our comparisons are made at fixed heights above ground, not relative to a shifting snow surface.

*How do you justify that 15 cm is a mid-snowpack position for a snowpack depth of 90 cm?*

We thank the reviewer for pointing this out. The intake heights were fixed relative to the ground (5 cm and 15 cm), and we mistakenly described the 15 cm level as "mid-snowpack." We agree this wording is misleading, particularly during periods when total snow depth exceeded 50–90 cm. In the revised manuscript we now consistently describe the intakes as "5 cm" and "15 cm above ground level within the snowpack." We no longer refer to them as "mid-snowpack," but rather emphasize that they represent lower snowpack pore-space levels, with overlying snow depth varying over time.

*I questioning the wind pumping effect on the isotope signal without knowing your spatial density profile of your snowpack. In addition, based on Figure 2 you measured a max. wind speed of 2 m/s and you are intakes are between 0.4m and 0.8m below the snowpack surface. For this condition I would not expect any significant ventilation inside the snow pack (see Colbeck et al., 1989). But without knowing your density profile of your snowpack it is hard to make a conclusion.*

We acknowledge that we did not measure a snow density profile, and this limitation is now stated explicitly in the manuscript. Nevertheless, as outlined in our response to the major comment, we carried out a diffusion–ventilation analysis. This analysis shows that barometric pressure oscillations routinely exceeded the thresholds required for sub-hourly exchange supports our interpretation that ventilation events did occur despite modest wind speeds.

**Specific Comments:**

**2) Methods**

*Line 115: Did you check your system for leaks? And did you do a humidity correction of the Picarro? It seems like that either your setup was not leak-tight or there is an offset in the humidity measurement of your Picarro. See major comments.*

Yes, we checked the system for leaks prior to deployment using a pressure-hold test, which confirmed that the sampling lines were leak-tight. We did not apply a humidity correction to the Picarro data; instead, we validated the raw humidity measurements against a co-located RH sensor. This comparison showed very good agreement ($R^2$ = 0.91, RMSE = 5.3 %RH, mean bias = −2.2 %RH), confirming that no systematic offset was present. Details are provided in our response to the major comment.

*Line 115: In Figure 2 you showed the snow depth. I would suggest to quickly mention it here with all the other parameters.*

Done

*Line 115: Is there a specific reason why you measured the isotopic composition at 5cm, 15cm and 1.5m above ground? I would suggest to provide a reason for it.*

The 5 cm inlet was deliberately planned to monitor pore vapor near the basal snow. The 15 cm inlet, however, was unintentionally buried deeper than intended during an early-season snowstorm and was not repositioned, in order to avoid disturbing the snowpack structure. We had already noted this point in the limitations section of the original manuscript. Finally, the 1.5 m inlet was placed in ambient air to characterize atmospheric vapor this height was selected because ~1 m corresponds to the average historical snow depth at the site, ensuring that the inlet remained above the snowpack throughout the season.

*Line 135-136: How did you make sure that the sensors did not get frozen during the measurements, especially your humidity sensor?*

The ambient RH/temperature sensor at 1.5 m was installed in a radiation shield, which prevented frosting and ensured reliable measurements. For the buried RH probes at 5 and 15 cm, no active heating was applied to avoid disturbing the snowpack structure. As a result, these probes were prone to frost deposition and often reported near-constant values around 100% RH. Hence, the in-snow RH measurements were likely affected by frost. In the original manuscript we stated that the snowpack was saturated based on these measurements; however, we have now removed this claim in the revised version

*Line 141-142: How did you make sure that the intakes are not sucking in small ice/water particles?*

The final 0.5 m of each intake line was placed horizontally within the snowpack and terminated in a downward-facing stainless-steel frit (1" diameter, 10 μm pore size). This configuration minimized the risk of entraining small ice particles into the sampling lines while preserving the integrity of the surrounding snow. The frit also introduced hydraulic resistance (detailed calculation in response to reviewer 3), confining the sampling footprint to the immediate vicinity of the inlet.

*Line 145-147: Could you provide a bit more explanation how you buried the last 0.5m inside the snow. Were the last 0.5m directly horizontally inside the snow? If not, how did you make sure that the temperature gradient inside the snowpack does not have an impact on condensation inside the tube?*

Yes, the final ~0.5 m of each intake line was buried directly and horizontally within the snowpack, maintaining thermal continuity with the surrounding snow. This segment was also insulated, which, together with the lack of active heating, minimized the risk of condensation inside the tube.

*Line 146: '... and minimizing the risk of condensation': It is hard to understand what you want to say here. I assume the reason why you didn't heat the last 0.5m is because you could have potential*

*heating up/melting of the snow at the intake. Why is there then a risk of condensation? Could you elaborate a bit more.*

You are correct, the last ~0.5 m of tubing was intentionally left unheated to avoid warming or melting the surrounding snow at the intake. The phrase "minimizing the risk of condensation" referred to the fact that this segment was buried horizontally, insulated, and maintained at the local snow temperature. We have rephrased the sentence in the Methods to read: "...the final 0.5 m of tubing was unheated but insulated and buried horizontally at intake height, ensuring thermal equilibrium with the surrounding snow and preventing condensation inside the line."

*Line 148-149: How far was the valve actuator away from the intakes?*

The valve actuator was housed at room temperature and located approximately 5.5 m from the inlets.

*Line 149: '... the three inactive sampling lines...' How can you have three inactive sampling lines when you have only three sampling lines. Why do you have a fourth one? Is it relevant to have a fourth one? Please provide more information.*

The system was designed with four measurement lines in total. The original purpose of the fourth line was to monitor vapor isotopic composition at <5 cm above the snowpack surface. However, as the season progressed, this inlet was repeatedly buried under new snowfall. To minimize complexity in the dataset we decided not to include this line in the present analysis.

**Methods**

*Line 149: '... continuously flushed': Please specific mention it that it continuously flush the line with the vapor of the pore space (I assume this is what you did).*

Yes, that is correct. The inactive sampling lines were continuously flushed with vapor from their respective pore-space locations. This ensured that fresh vapor was always present in the lines, reducing residence time and preventing stagnation before switching to active sampling. We have revised the Methods to specify that the continuous flushing was with pore-space vapor.

*Line 154: I don't see a reason why you should discharge the last 2 minutes before switching the valve. Did you see any impact on the signal in the last 2 minutes? If yes, why do you think that it has to do something with the valve? Could you elaborate a bit more?*

We did not observe any influence of valve switching but excluded the last two minutes to retain a central measurement window.

*Line 155-156: I would delete this sentences because it confuses a bit. Actually, you measured one intake twice during one hour.*

Our system included four lines, and the valve switched every 15 minutes, so each intake was measured once per hour.

*Line 158-161: I'm questioning your explanation about water percolation. Looking at Figure 2 I don't see a reason why you should have water percolation into the snowpack. The ambient temperature was far below zero and also the temperature at 15cm shows value below zero. Could it not be that the Picarro sucked in ice crystals which melted inthe heated tube? How do you prevent that ice particles can enter the intakes?*

As noted in the manuscript, we discarded all 15 cm measurements after 7 March 2023 because the Picarro issued a high-water concentration alarm. On that date, hourly averaged temperatures

exceeded 0 °C for ~6 hours, even though the daily boxplots in Figure 2 (of the manuscript) smooth over these excursions. This warm spell coincided with the instrument alarm, after which all data from the 15 cm line were excluded. Entry of ice particles was unlikely due to the intake design (downward-facing stainless-steel frit, 10 μm pores, final 0.5 m of tubing buried horizontally and insulated).

*Line 170: 'Humidity corrections were applied...': I assume it is not a humidity correction but a humidity-isotope correction.*

Yes, thank you for pointing this out. We have corrected the wording to "humidity–isotope correction" in the revised manuscript to avoid confusion.

*Line 177: '... minimal drift over time': At what humidity level did you perform your drift measurement? Close to the humidity levels of the intakes inside the snowpack?*

Thank you for this question. The drift measurement was performed at ~20,000 ppmv, i.e. at higher humidity levels than those inside the snowpack. This was done to ensure stable instrument operation and ideal conditions for evaluating long-term drift.

*Line 181-182: You indicate that 10% of your vapor concentration measurements inside the snow were below 2000 ppmv. I doubt that this is realistic for a snowpack where the lowest temperature was around -4 degrees. Could you check what was the snowpack temperature when you measured vapor concentration below 2000 ppmv.*

We checked all cases with vapor concentrations <2000 ppmv. At 5 cm, these occurred when the snowpack temperature was between about −1.7 and −0.2 °C (median −1.5 °C early period ; −0.26 °C late period). At 15 cm, they occurred when the snowpack was colder, between about −4.7 and −0.7 °C (median −4.4 °C early period; −1.05 °C late period). At the same times, the ambient air was much colder (−19 to −14 °C). As noted in our major comment response, this indicates that undersaturation arose from mixing with colder, drier ambient air rather than from measurement artifacts.

*Line 195-196: Could you elaborate a bit more why you didn't include sample measuring above 20 cm from ground level? Such measurements would help to get a better idea how the signal changed with depth and why there was a potential disequilibrium at the intakes.*

The 5 cm inlet was deliberately planned to monitor pore vapor near the basal snow. The 15 cm inlet, however, was unintentionally buried deeper than intended during an early-season snowstorm and was not repositioned, in order to avoid disturbing the snowpack structure. Finally, the 1.5 m inlet was placed in ambient air to characterize atmospheric vapor; this height was selected because ~1 m corresponds to the average max historical snow depth at the site, ensuring that the inlet remained above the snowpack throughout the season. We had already noted the burial of the 15 cm inlet in the limitations section of the original manuscript, and we now clarify these points in the Methods as well.

*Line 202: '... using a Picarro CRDS system.' Could you mention which model e.g. L2120, L2130 or L2140?*

Picarro L2130-i

*Line 216: I assume that the relative humidity was measured at 1.5 m. Please mention it.*

Corrected

*Line 231: 'Relative humidity within the snowpack remained saturated ...': I can't find any evidence that your data shows saturated conditions inside the snowpack. See comments above.*
We have removed this claim as explained in response to major comment.

**3) Results**

*Figure 2: How did you measure the snow depth and what was the resolution of it?*

Snow depth at the site was initially monitored using daily manual readings from a snow stake. These measurements generally agreed within 1–2 cm of the automated observations from Ted Stevens Anchorage International Airport (~7 km away). However, because the manual record was occasionally missed or became inconsistent during periods of heavy snowfall, we ultimately relied on the continuous 6-hourly snow depth data from the airport station. We have clarified this in the revised Methods section.

*Figure 3: Could you also add the temperature profile to see whether the concentration (ppmv) is following the temperature profile or not.*

We have revised Figure 3 (shown below) to include the temperature profiles at 5 cm, 15 cm, and 1.5 m alongside the corresponding water vapor concentrations and isotope compositions. The new temperature panels allow direct comparison of concentration (ppmv) with co-located snowpack and atmospheric temperatures. At 1.5 m, vapor concentrations broadly covary with temperature, consistent with Clausius–Clapeyron control on ambient air humidity. By contrast, at 5 cm and 15 cm within the snowpack, vapor concentrations show weaker and more variable correspondence with local temperatures, reflecting the additional influence of vapor diffusion, soil vapor input at the base, and ventilation processes. We believe this addition clarifies the relationship between vapor concentration and temperature, and strengthens the interpretation of undersaturation and non-equilibrium dynamics within the snowpack.

[Figure]

*Figure 3: Is it possible to extract a time-shift between the measured atmosphere data and inside the two snowpack locations? If yes, would it be possible to compare this time-shift with the diffusion time ($\Delta t = L^2/D$) from the snowpack surface to the intake locations (maybe include a tortuosity factor for the diffusion length inside the snowpack) to check whether it is consistent.*

We extracted the time-shift by cross-correlation between ambient vapor (1.5 m) and the pore-space vapor at 15 cm and 5 cm depth. In all continuous windows, the maximum correlation occurred at zero lag on the hourly series. Because our multiplexed sampling is based on 15-min intervals (later averaged to hourly), this means that the pore-space vapor responded essentially synchronously with the atmosphere, with an upper bound of ≤ 1 h for any lag. We then compared this observational constraint to diffusion timescales that include a tortuosity correction (Figure 2) . For the 15 cm intake, diffusion times are in the range 3–7 h, while for the 5 cm intake they are 5–10+ h. Thus, molecular diffusion is far too slow to explain the observed near-synchronous (< 1 h) coupling between atmosphere and pore-space vapor. Instead, the comparison highlights that ventilation processes (barometric pumping), which operate at sub-hourly scales, are required to transmit the atmospheric signal into the snowpack.

*Figure 4: I would suggest to change the colour of the '0-5 cm' and 5-10 cm' snow data points. It is hard to distinguish it.*

We have changed the colour in the revised manuscript

*Figure S4: How do you explain that you measured a vapor concentration below 1900 ppmv inside the snow pack but your temperature is only around -4 degrees? I'm also surprised that the vapor concentration inside the snowpack is almost the same as the 1.5m measurement.*

We examined all cases where in-snow vapor concentrations dropped below 1900 ppmv. At 15 cm depth, these occurred when snowpack temperatures were −4.7 to −3.9 °C (median −4.4 °C). At such temperatures, saturation would correspond to ~4300 ppmv, indicating clear undersaturation relative to local ice. At the same times, the ambient air was much colder (−19 to −15 °C) and contained only 1400–1800 ppmv vapor. The overlap between in-snow and ambient concentrations therefore reflects exchange with colder, drier air. As outlined in our major comment response, ventilation by barometric oscillations and episodic wind pumping can replace pore air on sub-hourly timescales, faster than sublimation can restore vapor to saturation. This mechanism explains why in-snow vapor concentrations could drop well below the ice-saturation value at the measured snow temperature. More generally, however, in-snow vapor concentrations were higher than those in the overlying atmosphere. Across the full record, median vapor concentrations were 4309 ppmv at 5 cm and 4441 ppmv at 15 cm, compared to 3496 ppmv in ambient air. Thus, while short episodes of ambient–snowpack similarity occurred during ventilation events, the broader dataset shows that pore-space vapor was usually enriched relative to the atmosphere.

**4) Discussion**

*Line 459-460: '... at different snowpack heights (5 cm, 15 cm, and 1.5 m)...' -> please rewrite this part because 1.5m does not belong to the snowpack but to the atmosphere.*

Thanks, this has been corrected

*Line 487-488: '.. than in ambient air, Within the ...': I assume that the sentence ends after 'ambient air.'*

Corrected

*Line 527: I questioning this paragraph without knowing your spatial density profile of your snowpack. In addition, based on Figure 2 you measured a max. wind speed of 2 m/s and you are intakes are between 0.4m and 0.8m below the snowpack surface. For this condition I would not expect any significant ventilation inside the snow pack (see Colbeck et al., 1989). But without knowing your density profile of your snowpack it is hard to make a conclusion. I would suggest that you provide more evidence to support your hypothesis. Maybe you could provide an estimation about what wind speed inside the snowpack would be needed to transport the atmospheric vapor into the snowpack. E.g you could try to extract a time-shift between the measured atmosphere data and inside the two snowpack locations and calculate a wind speed needed to transport the signal into the snowpack.*

We agree that without a detailed density–depth profile it is impossible to compute a precise wind-driven ventilation rate. For this reason we did not attempt to estimate wind-driven advection in the manuscript. Instead, we adopted a wide range of snowpack permeabilities, porosities and tortuosities from the literature and showed that the pressure difference required to ventilate 0.05 and 0.15 m heights above the ground in 15–60 min windows is small ($\approx 10^{-2}$ to $10^{1}$ Pa) compared to the barometric pressure variations measured in our study. Wind pumping occasionally reaches the threshold, but it is barometric oscillations, operating independently of wind speed, that routinely exceed it. Figure 3 illustrates this magnitude comparison, while Figure 4 provides a categorical time-series view: for the "average snowpack" case, barometric forcing alone exceeds the required threshold almost continuously, whereas wind forcing does so only sporadically. Thus, ventilation can occur on sub-hourly time scales even with weak ambient winds because pressure oscillations propagate through the snowpack and drive advective flow

*Line 608: '... in the mid-snowpack (around 10-15 cm).': How do you justify that 15 cm is a mid-snowpack position for a snowpack depth of 90 cm?*

We thank the reviewer for pointing this out. The intake heights were fixed relative to the ground (5 cm and 15 cm), and we mistakenly described the 15 cm level as "mid-snowpack." We agree this wording is misleading. In the revised manuscript we now consistently describe the intakes as "5 cm" and "15 cm above ground level within the snowpack.

*Line 606 and 610: Would it be possible to provide an explanation that first 'In early winter, the snowpack behaved as a closed system...' and afterwards the snowpack is not closed anymore and wind-pumping and '... ventilation became the primary transport mode...'? Looking at your snow depth data on Figure 2 I would expect that your two intakes locations inside the snowpack are even more decoupled from the atmosphere because the snowpack is rising by additional 20-30cm.*

In our original submission we described the system as transitioning from "closed" to "open." Based on further analysis of barometric pressure fluctuations, we now recognize that barometric pumping was active throughout the record and consistently ventilated the pore space. To avoid the misleading impression that the snowpack was ever fully closed, we revised the text to frame the seasonal evolution as a shift in the relative dominance of diffusion versus ventilation.

In the early period, strong soil–snow thermal gradients sustained upward diffusion, producing clear isotopic stratification: vapor near 5 cm was relatively enriched in $\delta^{18}O$ (–33.7‰) with low d-excess (6.7‰), whereas vapor at 15 cm was more depleted (–36.3‰) with elevated d-excess (17.5‰). Although barometric pumping was present, its isotopic imprint was muted because ambient vapor (–33.5‰, 8.6‰) was similar to pore vapor near the base, so ventilation did not override the diffusion signal.

After the 24 January warm event, thermal gradients collapsed and diffusion weakened, while isotopic contrast with ambient vapor grew. For example, in the mid period, ambient vapor was enriched (–29.8‰, 6.9‰) relative to pore vapor (5 cm: –33.5‰ / 9.3‰; 15 cm: –34.0‰ / 10.4‰). By the late period, pore vapor values converged toward ambient (5 cm: –33.7‰ / 7.5‰; 15 cm: –32.9‰ / 6.2‰; ambient: –31.9‰ / 6.7‰), vertical stratification diminished, and diurnal cycles emerged. These shifts show that while ventilation was always active, its isotopic imprint became dominant only once ambient vapor diverged from the internal reservoir. Importantly, even with a 20–30 cm increase in snow depth, pore spaces were not decoupled from the atmosphere, as reflected in convergence of $\delta^{18}O$ and d-excess toward ambient values and the emergence of diurnal cycles.

*Line 636-637: Could you elaborate this a bit more? What do you mean '... buried well below its intended mid-snowpack position during an early-season snowstorm'? Didn't you want to keep the intake locations constant at 5 cm and 15 cm above ground? Or what was your intended mid-snowpack position? And how do you justify that 15 cm is a mid-snowpack position for a snowpack depth of 90 cm?*

We appreciate the reviewer's request for clarification. The 5 cm inlet was deliberately positioned to monitor pore vapor close to the basal snow. The 15 cm inlet was installed above it, but during an early-season snowstorm it was buried more deeply than originally intended and was not repositioned in order to preserve snowpack stratigraphy. We recognize that describing this level as "mid-snowpack" was misleading, particularly once snow depth exceeded 50–90 cm. In the revised manuscript we now consistently describe the intakes as being fixed at "5 cm" and "15 cm above ground level within the snowpack," rather than referring to them as "mid-snowpack." We have clarified these points in the Methods and Limitations sections

**5) Conclusion**

*Line 692: The link is not working. Please correct it.*

Thank you for flagging this. The dataset has been uploaded to our repository and is currently under restricted (embargoed) access for peer review. In line with our policy, the archive will be made public upon publication (DOI listed in the Data Availability statement). If the reviewer would like to inspect the data during review, we will be happy to provide temporary private access upon request.

*S1) Supplement material*

*Line 4: '... fluctuations in water vapor com concentrations.' -> remove 'com'*

Done

*Line 7: '... relative humidity at the four inlets, a HOBO...': I think there is a type. You are talking about four inlets but based on your experimental setup you have only three (5cm, 15cm and 1.5m).*

As explained previously, the system was designed with four measurement lines in total. The original purpose of the fourth line was to monitor vapor isotopic composition at <5 cm above the snowpack surface. However, as the season progressed, this inlet was repeatedly buried under new snowfall. To minimize complexity in the dataset and maintain consistency across levels, we decided not to include this line in the present analysis.

**Technical comments:**

*Line 255: '... the snowpack slightly enriched than the ...' Typo -> '... the snowpack was slightly more enriched than the ...'*

Corrected

*Line 339: 'The 5-10 cm layer shows a steeper slope ...' Typo -> 'The 5-10 cm layer showed a steeper slope ...'*

corrected

*Line 354: '... vapor was negatively correlated ...' Typo -> '... vapor were negatively correlated ...'*

corrected

*Line 678: 'Our data set ...' -> 'Our dataset ...'*

corrected

*Remove redundant commas in citations (e.g., "Bailey et al., (2019)" → "Bailey et al. (2019)")*

done

**APPENDIX:**

**Diffusion Timescales Pressure Gradient Requirements and Atmospheric Forcing**

**1. Diffusion timescales (t_diff)**
To provide a baseline for snowpack air exchange in the absence of pressure forcing, we calculated the characteristic diffusion timescale:

$t\_diff = L^2 \div D\_eff,$

where L is the distance from the intake to the snow surface (0.05 m or 0.15 m), Because L increases as the snowpack deepens, diffusion timescales lengthen markedly through the season. D_eff is the effective diffusivity of water vapor through snow. D_eff was calculated as:

$D\_eff = D\_air \div \tau,$

where D_air is the molecular diffusivity of water vapor in air ($2.1 \times 10^{-5}$ $m^2$ $s^{-1}$) and $\tau$ is the tortuosity factor (tested over 1.1–1.4).

**2. Required pressure gradients (ΔP_req)**
We estimated the pressure difference required to ventilate pore air in the snowpack by combining prescribed lag times with Darcy's law. The lag time is not an observed variable, but a benchmark that we imposed to test how much pressure forcing would be required for ventilation at different timescales. Conceptually, the question is: *If air must be exchanged between the snow surface and a depth of 5 cm or 15 cm within a specified lag time, what pressure difference is needed to achieve that exchange?*

First, the lag time was converted into a pore velocity:

- Pore velocity (v_p) = exchange depth (L) ÷ lag time (s).

This pore velocity represents the effective air movement that would be needed to flush the pore space within the specified time window.

Darcy's law relates this pore velocity to the required pressure difference across the snow depth:

- $\Delta P = (\mu \times \phi \times v\_p \times L) \div k$.

$\mu$ is the dynamic viscosity of air ($1.7 \times 10^{-5}$ Pa s), $\phi$ is the porosity of the snow, $v\_p$ is the pore velocity defined above, L is the vertical distance over which air must move (0.05 m for the 5 cm sensor level; 0.15 m for the 15 cm sensor level), k is the permeability of the snow.

We explored a range of realistic seasonal snow properties: Permeability (k): $10^{-11}$ to $10^{-7}$ m$^2$, Porosity ($\phi$): 0.55 to 0.80 (Calonne et al., 2012). Tortuosity ($\tau$): 1.1 to 1.4, reflecting the meandering of flow paths through the snow matrix (Lieblappen et al., 2020)

For each lag window (15, 30, and 60 minutes), $\Delta P\_req$ was calculated across all combinations of k, $\phi$, and $\tau$. The resulting shaded bands in the plots show the minimum–maximum envelope of $\Delta P\_req$, while the solid (5 cm) and dashed (15 cm) curves represent the geometric mean values.

**3. Observed atmospheric forcings**
We then compared the calculated $\Delta P\_req$ against two atmospheric mechanisms capable of generating pressure gradients in snow: wind forcing and barometric forcing. Both forcings were aggregated into 15, 30, and 60 minute sliding windows to directly match the lag times used for $\Delta P\_req$.

- Wind forcing (dynamic pressure):
  Dynamic pressure was calculated as
  $\Delta P\_wind = \frac{1}{2} \times \rho \times U^2$,
  where $\rho$ is air density (1.3 kg m$^{-3}$) and U is the maximum wind speed observed within the lag window. Dynamic pressure represents the potential overpressure available from wind pumping during gusts.

- Barometric forcing (pressure oscillations):
  Barometric pressure forcing was defined as the absolute difference between maximum and minimum atmospheric pressure within each lag window:
  $\Delta P\_baro = |P\_max – P\_min|$.
  This captures oscillatory barometric pumping of the snowpack and was calculated using pressure measurements spanning a 30 m vertical offset.

The observed forcings and the required $\Delta P$ were plotted on the same axes to allow a direct visual comparison of magnitudes and timescales.

**References:**

Bouvet, L., Calonne, N., Flin, F., and Geindreau, C.: Multiscale modeling of heat and mass transfer in dry snow: influence of the condensation coefficient and comparison with experiments, The Cryosphere, 18, 4285–4313, https://doi.org/10.5194/tc-18-4285-2024, 2024.

Calonne, N., Geindreau, C., Flin, F., Morin, S., Lesaffre, B., Rolland du Roscoat, S., and Charrier, P.: 3-D image-based numerical computations of snow permeability: links to specific surface area, density, and microstructural anisotropy, The Cryosphere, 6, 939–951, https://doi.org/10.5194/tc-6-939-2012, 2012.

Ebner, P. P., Steen-Larsen, H. C., Stenni, B., Schneebeli, M., and Steinfeld, A.: Experimental observation of transient $\delta^{18}O$ interaction between snow and advective airflow under various temperature gradient conditions, The Cryosphere, 11, 1733–1743, https://doi.org/10.5194/tc-11-1733-2017, 2017.

Graham, L. and Risk, D.: Explaining $CO_2$ fluctuations observed in snowpacks, Biogeosciences, 15, 847–859, https://doi.org/10.5194/bg-15-847-2018, 2018.

Lieblappen R, Fegyveresi JM, Courville Z and Albert DG, Using Ultrasonic Waves to Determine the Microstructure of Snow. Front. Earth Sci. 8:34. doi: 10.3389/feart.2020.00034, 2020

Massman, W. J., Sommerfeld, R. A., Mosier, A. R., Zeller, K. F., Hehn, T. J., & Rochelle, S. G. (1997). A model investigation of turbulence-driven pressure-pumping effects on the rate of diffusion of CO2, N2O, and CH4 through layered snowpacks. *Journal of Geophysical Research: Atmospheres*, *102*(D15), 18851-18863.

Massman, W., Sommerfeld, R., Zeller, K., Hehn, T., Hudnell, L., & Rochelle, S. (1995). CO2 flux through a Wyoming seasonal snowpack: diffusional and pressure pumping effects. In *In: Tonnessen, Kathy A.; Williams, Mark W.; Tranter, Martyn, eds. Biogeochemistry of Seasonally Snow-Covered Catchments (Proceedings of a Boulder Symposium, July 1995). IAHS Publ. No. 228. International Association of Hydrological Sciences. p. 71-79.* (pp. 71-79).